# COMPOSABLE INTERVENTIONS FOR LANGUAGE MODELS

**Arinbjörn Kolbeinsson***
University of Virginia & Askan
arinbjorn@virginia.edu

**Kyle O'Brien***
EleutherAI

**Tianjin Huang***
University of Exeter

**Shanghua Gao**
Harvard Medical School

**Shiwei Liu**
University of Oxford

**Jonathan Richard Schwarz**
Thomson-Reuters
Foundational Research

**Anurag Vaidya**
Harvard Medical School
Mass General Brigham

**Faisal Mahmood**
Harvard Medical School
Mass General Brigham

**Marinka Zitnik**
Harvard Medical School

**Tianlong Chen**
UNC Chapel Hill

**Tom Hartvigsen**
University of Virginia & Thomson-Reuters Foundational Research
hartvigsen@virginia.edu

## ABSTRACT

Test-time interventions for language models can enhance factual accuracy, mitigate harmful outputs, and improve model efficiency without costly retraining. But despite a flood of new methods, different types of interventions are largely developing independently. In practice, multiple interventions must be applied sequentially to the same model, yet we lack standardized ways to study how interventions interact. We fill this gap by introducing *composable interventions*, a framework to study the effects of using multiple interventions on the same language models, featuring new metrics and a unified codebase. Using our framework, we conduct extensive experiments and compose popular methods from three emerging intervention categories—*knowledge editing*, *model compression*, and *machine unlearning*. Our results over 417 different compositions uncover meaningful interactions: compression hinders editing and unlearning, composing interventions hinges on their order of application, and popular general-purpose metrics are inadequate for assessing composability. Taken together, our findings showcase clear gaps in composability, suggesting a need for new multi-objective interventions.[1]

## 1 INTRODUCTION

Language models (LMs) exhibit striking capabilities on important tasks in many domains including medicine (Singhal et al., 2023), finance (Wu et al., 2023), science (Taylor et al., 2022), and entertainment (Zhong et al., 2023a). But despite high performance, deployed LMs can still misbehave unpredictably and require updates. For example, LMs generate content that is *hallucinatory* (Ji et al., 2023), *factually incorrect* (Zhao et al., 2023b), and *harmful* (Mendelsohn et al., 2023; Jain et al., 2024; Hartvigsen et al., 2022). Beyond unwanted behaviors, user requirements also change over time. For example, regulations arise (The President of the United States, 2023), computational resources constrict, knowledge gets outdated (Tack et al., 2024), and copyrighted training

---

[1]All of our code is available at: github.com/hartvigsen-group/composable-interventions
*Equal contribution.

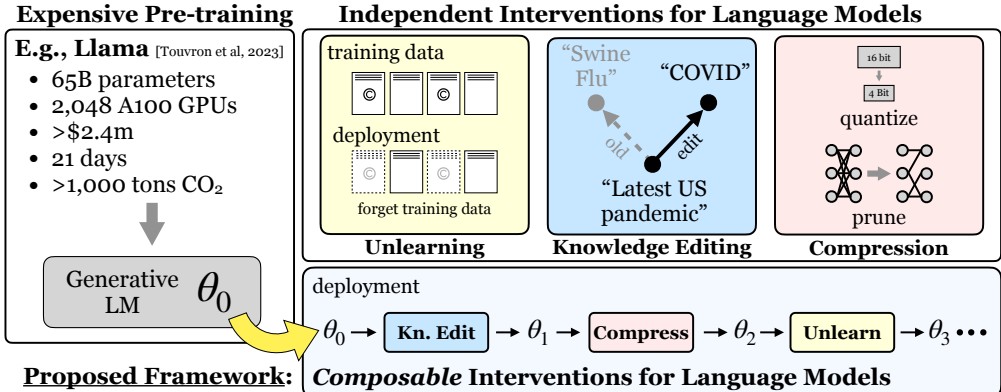

Figure 1: Interventions aim to update targeted properties of language models without impacting unrelated behaviors or adding excessive compute. We introduce and extensively experiment with *composability* of different interventions used on the same model.

materials are identified (Grynbaum & Mac, 2023). Without ways to quickly address these issues, models can be left unsafe, outdated, biased, and non-compliant with laws or regulations, limiting their widespread responsible use (Kaddour et al., 2023).

Many recent works study efficient, in-place updates for LMs. We broadly refer to these as *interventions*—updates to targeted properties of LMs applied after pretraining (and optional fine-tuning). For example, we can view model compression (Zhu et al., 2023; Frantar et al., 2023; Frantar & Alistarh, 2023) as an intervention to make language models more inference- or memory-efficient. Other rapidly-advancing examples include knowledge editing (Mazzia et al., 2023; Hartvigsen et al., 2023; Meng et al., 2022; 2023; Yu et al., 2024; Tan et al., 2024), detoxification (Welbl et al., 2021; Yu et al., 2023; Wang et al., 2022; Suau et al.), and unlearning (Liu et al., 2024a; Lynch et al., 2024; Eldan & Russinovich, 2023; Lucki et al., 2024). However, these interventions are largely advancing independently. In practice, we usually have *multiple* requirements for properties of our models (*e.g.,* factuality *and* efficiency). And as requirements change over time, new interventions must be applied. While some works have started studying interactions between training objectives (Matzken et al., 2023; Xu et al., 2023b; Li et al., 2024b), practical widespread use is limited without unified evaluations for how interventions interact.

As depicted in Figure 1, we propose that practical interventions should be *composable*: When an intervention is applied to a model, it should avoid interfering with success of prior or future interventions. For example, a model user might need to repair a factual error in an LM using a knowledge editing method. Later on, they might have to quantize that model for deployment on mobile devices. A *composable* quantization method should specifically preserve the edited fact and a composable editor should withstand quantization. However, it remains unknown how well existing interventions compose, and we lack formal notions of intervention composition. We therefore propose two metrics for composability: 1) *Order-free Error*, where an intervention is composable if its application leaves others' success unimpacted, and 2) *Order Sensitivity*, where the combined success of multiple interventions should not depend on the order in which they are applied. We instantiate these metrics in a codebase that includes popular interventions, tackling a major lack of standardized code that has inhibited cross-intervention evaluations until now.

We use our framework to extensively study composability of state-of-the-art *knowledge editing*, *model compression*, and *machine unlearning* interventions on popular and recent LMs. Our experiments with 417 different compositions unearth novel insights and offer guidance for composing recent methods. Three key results are as follows: ① **Model compression often limits the success of other interventions**. This suggests a significant drawback of existing compression methods, which are nearly universal in LM deployments. ② **The order in which interventions are applied dramatically alters their success**. Therefore, we need new interventions explicitly designed for

composability. ③ **General-purpose post-intervention model performance is a poor proxy for composability**. This indicates that targeting composability as a metric has the potential to drive the development of new, practically-grounded interventions. Taken alongside our other findings (Section 4), our experiments suggest we sorely need to broaden intervention method evaluations and to design new methods explicitly for composability.

Our key contributions are as follows:

- We introduce the notion of *composability* for post-training interventions to LMs. Our work opens doors to address crucial, practical challenges in online LM updates and broadens evaluation criteria for interventions.

- Our main contribution is extensive experimentation, identifying unknown interactions between knowledge editing, model compression, and machine unlearning.

- Our findings suggest a clear need to develop novel interventions that target composability, a crucial property of practical interventions.

- We release an extendable codebase that unifies many state-of-the-art implementations to enable others to develop new multi-objective interventions.

## 2  METHODS TO EVALUATE INTERVENTION COMPOSABILITY

### 2.1  PRELIMINARIES: INTERVENTIONS FOR LANGUAGE MODELS

Assume we are given a language model $f_\theta$, where $\theta \in \Theta$ contains its pre-trained parameters and $\Theta$ is the parameter space. We define an *intervention* $\phi$ as the combination of four things: 1) an operator $\omega(f_\theta, \gamma)$ that produces a new model $f_\theta$, 2) hyperparameters $\gamma$, which typically control the strength of intervention, 3) a loss $\ell(\omega(f_\theta, \gamma))$ that specifies the targeted update,[2] and 4) evaluation criteria $\kappa(f_\theta, \mathcal{D}) \in [0, 1]$, where higher scores indicate better performance. Each intervention's operator $\omega$ aims to minimize loss $\ell$ and ultimately achieve high performance according to $\kappa$. For readability, we denote the application of intervention $\phi$ to model $f_\theta$ as $\phi(f)$.

### 2.2  COMPOSABLE INTERVENTIONS FOR LANGUAGE MODELS

We propose that multiple interventions should compose with one another. Consider $N$ possible interventions $\{\phi_i\}_{i=0}^{N-1}$. Intervention *composition* is the application of multiple interventions to a model $f$ in sequence.[3] For example, we might apply intervention $\phi_0$ to model $f_\theta$, then $\phi_1$ to the intervened-upon model. We can write this composition as either $\phi_1(\phi_0(f_\theta))$ or $\phi_1 \circ \phi_0(f_\theta)$.

Most interventions $\phi_i$ are developed specifically for their own criterion $\kappa_i$. For simplicity, we let each intervention have one criterion, though in practice many have multiple. We propose that the quality of many interventions in practice should also relate to *how an intervention impacts our ability to apply other interventions*. To meet this need, we propose two metrics that measure such *composability*. Each is computed for a given criterion $\kappa_i$ and can be computed for a single hyperparameter setting or ranged over multiple settings to attain a richer description of intervention interactions.

Our evaluation strategy is straightforward: After applying an intervention $\phi_i(f_\theta)$, we measure its success as $\kappa_i(\phi_i(f_\theta))$. Then, we can add another intervention $\phi_j$ and measure $\kappa_i(\phi_j \circ \phi_i(f_\theta))$. The difference between these measures then elicits how $\phi_j$ impacts the success of $\phi_i$ according to $\kappa_i$. To disentangle absolute performance on $\kappa_i$ from the interventions, we explore the impact of *ordering* by reversing the interventions and computing $\kappa_i(\phi_i \circ \phi_j(f_\theta))$. By pairing absolute changes with results from different orderings, we isolate direct interactions between interventions. Overall, this approach provides guidance when choosing interventions to compose and illuminates failure modes

---

[2]Some interventions require data, but we remove data-dependent notation for simplicity.
[3]For readability, we describe our framework in terms of two interventions, even though it scales.

of existing methods. We next describe two concrete composability metrics. The first describes best-case composability regardless of order, and the second isolates the impact of ordering.

**Composability Error Metric 1: Order-free Error.** Interventions are more composable if they achieve high performance when applied in either order. We therefore measure the impacts of adding a second intervention to a model regardless of order for a chosen criterion $\kappa_i$ using *order-free error*, computed as the area *above* the maximum of two curves. A small area means higher overall performance on the metric. We use the maximum criterion value as an upper bound for this area, and assume it is 1, though this is easily changed in practice. Each curve corresponds to an order in which each intervention is applied and sweeps over hyperparameter choices for $\gamma_i$:

$$1 - \int_{\gamma_i} \min(\underbrace{\kappa_i(\phi_j \circ \phi_i(f_\theta, \gamma_i))}_{\text{intervention order } i \to j}, \underbrace{\kappa_i(\phi_i(\gamma_i) \circ \phi_j(f_\theta))}_{\text{intervention order } j \to i}). \tag{1}$$

A smaller area indicates better composability. By sweeping over hyperparameters, this measure provides a general sense of composability without choosing hyperparameters ahead of time. The area can be computed for any criteria $\kappa$—in our experiments, we evaluate all criteria for each composition. In practice, not all hyperparameters are practical. For example, harsh model compression can have catastrophic impacts on model utility and so comparing interventions at extreme levels may be unrealistic. Therefore, we can also set a bound on acceptable performance decay with respect to $\kappa_i$ or any other criterion when using Equation 1 and can also easily be measured for hand-chosen hyperparameters, and for a single setting $\gamma_i$, Order-free Error reduces to the simple error score: $1 - \min(\kappa_i(\phi_j \circ \phi_i(f_\theta, \gamma_i)), \kappa_i(\phi_i(\gamma_i) \circ \phi_j(f_\theta)))$.

**Composability Error Metric 2: Order Sensitivity.** Interventions are more composable if their performance is invariant to their order of application. We measure *order invariance* for criterion $\kappa_i$ as the area *between* the two curves defined by each direction of intervention application. Again, we compute this across hyperparameter choices for $\gamma_i$:

$$\int_{\gamma_i} |\kappa_i(\phi_j \circ \phi_i(f_\theta, \gamma_i)) - \kappa_i(\phi_i(\gamma_i) \circ \phi_j(f_\theta))|. \tag{2}$$

A smaller area again indicates better composability. And again, if hyperparameters are already chosen, this simplifies to a straightforward error: $|\kappa_i(\phi_j(\phi_i(f_\theta))) - \kappa_i(\phi_i(\gamma_i) \circ \phi_j(f_\theta))|$.

Overall, measuring Order-free Error and Order Sensitivity provides the first insights into intervention composability. Each reduces to a simple form for point measures of the impact of composition. While more computationally costly, the broader sweeps over hyperparameters enable clearer pictures of how interventions interact. As shown in Section 3, the curves produced by these sweeps also enable intuitive, visual understandings of composability.

## 3 EXPERIMENTAL SETUP

We study the effects of sequentially applying multiple interventions to the same model on performance as measured by intervention-specific metrics and overall utility. Understanding these effects is crucial for the practical application of test-time interventions. The following sections detail our experiments to illuminate how composable various interventions are across model editing, unlearning, and compression.

### 3.1 INTERVENTION DATASETS, METHODS, AND METRICS

We select ten methods from the three categories of interventions below. Each features task-specific and overall utility metrics. Appendix B further elaborates and Table 1 summarizes our setup.

**Knowledge Editing.** LMs' knowledge can be fundamentally incorrect or grow outdated. Knowledge editing aims to update *facts* according to LMs, typically framed as question–answering. Here,

| INTERVENTIONS | METHODS | DATASETS | INTERVENTION METRICS | COMPOSABILITY METRICS | MODELS |
|---|---|---|---|---|---|
| **Knowledge Editing** | Finetuning, LoRA , MEMIT | zsRE | Edit Success, Edit Generalization, Strict Edit Locality, MMLU | Order-free Error (Equation 1) | Llama-3 8B |
| **Model Compression** | *Pruning:* Wanda, SparseGPT *Quantization:* GPTQ, AWQ | – | MMLU | | Mistral 7B Instruct Yi 1.5 9B Chat |
| **Machine Unlearning** | Gradient Ascent (GA) , Gradient Difference (GD), Representation Misdirection Unlearning (RMU) | WMDP | WMDP, MMLU | Order Sensitivity (Equation 2) | |

Table 1: Summary of the methods, data, and metrics used in our experiments. We use Llama-3 8B as the benchmark model in our main results (Section 4) and conduct additional generalizability experiments on Mistral 7B Instruct and Yi 1.5 9B Chat. Composability metrics apply to all experiments.

where models are edited to return updated answers to questions by increasing the likelihood of the correct answers. We use three methods, including the popular MEMIT (Meng et al., 2023) editor, as well as LoRA (Hu et al., 2021) and finetuning, which have recently been shown to be surprisingly successful editors, despite their simplicity (Gangadhar & Stratos, 2024; Hsueh et al., 2024; Hua et al., 2024). We use the zsRE (Levy et al., 2017) dataset, which is a popular question-answering benchmark for knowledge editing.

To evaluate editors' performance, we use three standard metrics (Yao et al., 2023): *Edit success* measures whether the post-edit model successfully outputs the correct, edited response. *Edit generalization* measures how well the edit generalizes to other semantically-equivalent inputs. We compute generalization as the average edit success across a set of 10 holdout edit rephrasings. *Strict Edit locality* measures the impact of edits on unrelated inputs as the average success on random, unedited samples. All editing metrics are computed as the F1 score on the correct output logits as a strict measure of editing performance—intended outputs must be the most-likely tokens after editing. Each metric's highest value is 1.0.

**Machine Unlearning.** LMs can learn undesirable knowledge from their pretraining data, including information that is potentially dangerous (Li et al., 2024a) copyrighted (Karamolegkou et al., 2023; Meeus et al., 2024), toxic (Sheng et al., 2019; Gehman et al., 2020; Hartvigsen et al., 2022), or memorized (Carlini et al., 2018; Ippolito et al., 2022; Biderman et al., 2023; Prashanth et al., 2024). Machine unlearning methods aim to remove undesirable knowledge from LMs without regressing performance on extraneous domains (Cao & Yang, 2015; Liu et al., 2024a). We experiment with three popular and representative methods: Gradient Ascent (GA) (Jang et al., 2022), Gradient Difference (GD) (Maini et al., 2024), and Representation Misdirection for Unlearning (RMU) (Li et al., 2024a). GA is a traditional approach that minimizes the likelihood of chosen unlearning targets. GD augments GA by also maintaining the LM's predictions on unrelated retention data. RMU, the most recent method, perturbs the model's representations for inputs related to the learning target. Further details are in Appendix C.4.

We evaluate unlearning with Weapons of Mass Description Proxy (WMDP) (Li et al., 2024a), updating LMs to forget potentially hazardous biosecurity and cybersecurity knowledge. We average the performance on WMDP's cyber and bio splits, totaling 3,260 questions. We report individual split performance in Appendix E. Optimal unlearning yields 25% accuracy as this is random performance on the four-choice benchmark.

Unlearning is a broad problem area. While, WMDP is among the most studied unlearning benchmarks, recent works have suggested that WMDP performance provides an imperfect evaluation of unlearning writ-large. We proceed with leveraging WMDP as it is a sufficient evaluation setting for measuring composability when applied to unlearning, but caveat that comprehensive evaluation of unlearning is an open research problem.

**Model Compression.** LMs require significant compute resources during pretraining, finetuning, and inference. Model compression methods aim to reduce resources needed at test-time. We experiment with four popular, state-of-the-art weight pruning and quantization methods. For pruning, we choose SparseGPT (Frantar & Alistarh, 2023) and Wanda (Sun et al., 2023). For quantization, we choose GPTQ (Frantar et al., 2023) and AWQ (Lin et al., 2023). Pruning and quantization are used for different reasons, so we generally avoid comparisons between the two. Each algorithm has a hyperparameter that controls the compression level—% of zeroed weights for pruning, *bits* for quantization. Naturally, efficiency grows with compression, but at the expense of task performance. We vary this hyperparameter to explore various levels of compression, choosing sparsity levels of .25, .35, .45, .55, .65, and .75 for pruning and 2, 3, 4, and 8 bits for quantization – down from 16 bits. We note that most compression techniques require decompressing models, so we recompress after editing or unlearning using the same compression technique.

**Measuring General Model Utility.** All interventions aim to avoid degrading overall model utility. While assessing general performance of LMs is an active area (Dehghani et al., 2021; Biderman et al., 2024; Gupta et al., 2024), we make the standard choice to evaluate question–answering accuracy on MMLU (Hendrycks et al., 2020) using the LM Eval Harness (Gao et al., 2023). Accuracy of 25% indicates random predictions.

### 3.2 Implementation details

All experiments in our main results (Section 4) are performed with Llama3-8B (AI@Meta, 2024), which is well-studied and highly capable for its parameter count. All results for knowledge editing methods are averaged over 10 batches of 50 randomly-selected edits from zsRE. Further details on implementations and hyperparameters are in Appendix C.

## 4 Results

In the following subsections, we first examine how each pair of interventions compose individually, then summarize general trends. We measure composability using our metrics from Section 2 for each intervention's criteria and report impacts of compression at different levels. All intervention pairs are applied in both directions (*e.g.,* MEMIT before AWQ and AWQ before MEMIT). For composability metrics, we compare methods regardless of metric ranges by counting the number of times each intervention method outperforms the others (# Wins) with ties counting as losses.

### 4.1 Composing Model Compression with Knowledge Editing

Our results are shown in Figure 2, where we find that model compression and knowledge editing interact in meaningful ways at different levels of compression. We report three editing metrics and MMLU to measure general model degradation. All results are averaged (mean) over 10 batches of 50 edits each. Illustrated below, these experiments reveal that there is a large gap in composability of existing methods. Our findings are as follows.

① **Model compression degrades editing performance.** Across Figure 2, edits made by MEMIT, LoRA, and Finetuning deteriorate as compression increases. In weight pruning, the editing metrics, Success, Generalization, and Locality, steadily drop with higher sparsity. For quantization, the steepest drops occur between 2 and 4 Bits, and GPTQ decays editing performance faster than AWQ. While general performance decay from compression is unsurprising, its steepness varies dramatically between editors and compression methods. Decay curves also depend on the order of interventions. Finetuning remains a surprisingly strong baseline at higher compression levels, often surpassing MEMIT or LoRA. Overall, these results suggests that model compression can limit a model's editability.

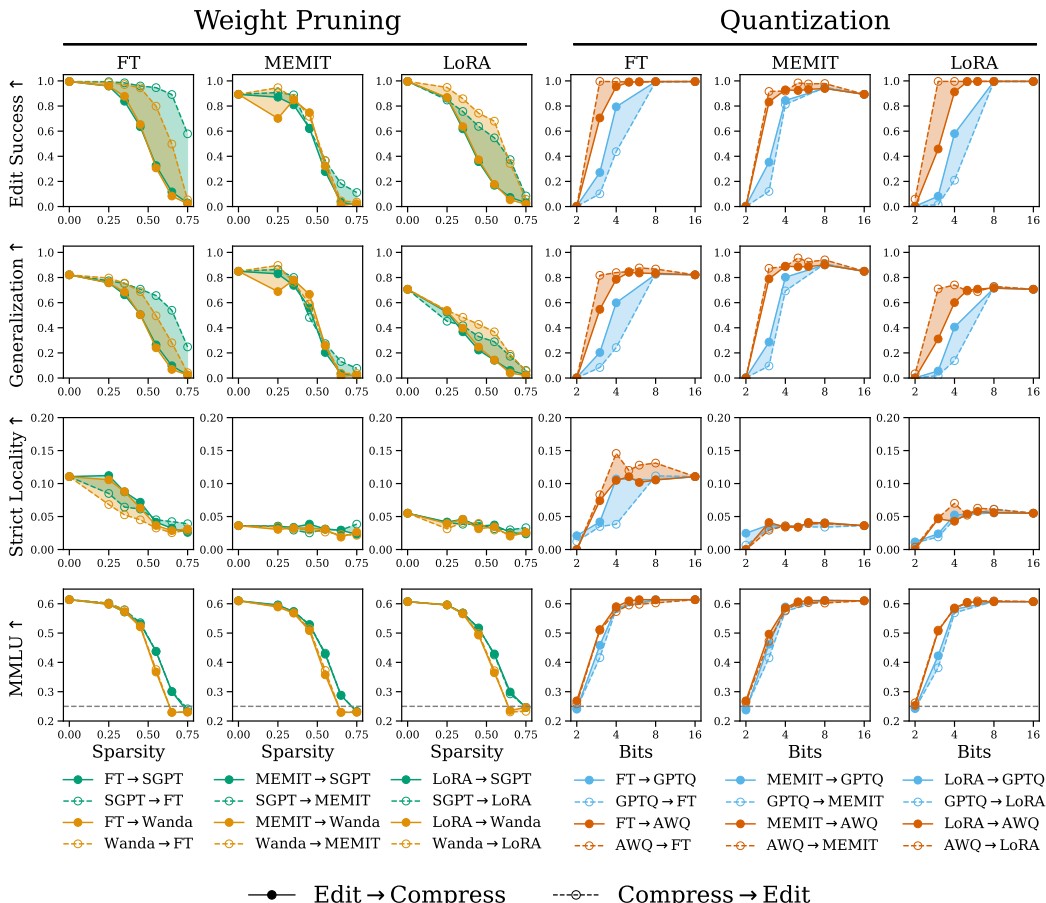

Figure 2: Composing Knowledge Editing with Model Compression with varying degrees of compression. Higher values are better for all metrics. **Key Takeaways:** Editing post-compression generally outperforms the reverse order, as compression tends to degrade edit performance and all editors exhibit order sensitivity.

② **Editing performance hinges on the order of interventions.** The shaded area between the curves in Figure 2 shows the impact of ordering. Ordering is highly impactful in most cases and varies dramatically between editing methods. For pruning methods (left side), compressing first is generally better than editing first. For example, pruning after Finetuning or LoRA destroys edits much faster than if the model had been compressed first. For quantization (right side), the order also matters, but the best order differs by method. For GPTQ, editing should be done first. For AWQ, compression should be done first. This finding suggests that these interventions interact in meaningful ways and successful composition will benefit from tailored, composable methods that have expected behaviors.

③ **Composability can vary within the same intervention category.** We next summarize the general composability of each intervention pair in Table 2, using the metrics from Section 2: Order-free Error (↓) measures best-case composition error, while Order Sensitivity (↓) measures unwanted effects of intervention ordering. These metrics let us compare methods according to composability, and we find that 1) MEMIT is generally the most composable editor, totaling 14 wins compared to 9 by Finetuning and 4 by LoRA, 2) SparseGPT is the most composable pruning method with 8 wins compared to Wanda's 6, and 3) AWQ far surpasses GPTQ, achieving 17 wins compared to GPTQ's 2. However, many wins are quite close and composability can change dramatically between metrics. These results indicate a need to use many metrics to explicitly target and evaluate composability.

| | Edit Success | | | | | | Edit Generalization | | | | | | |
| | Order-free Error (↓) | | | Order Sensitivity (↓) | | | Order-free Error (↓) | | | Order Sensitivity (↓) | | | |
| Method | FT | MEMIT | LoRA | FT | MEMIT | LoRA | FT | MEMIT | LoRA | FT | MEMIT | LoRA | # Wins |
|---|---|---|---|---|---|---|---|---|---|---|---|---|---|
| **Wanda** | .01 | .05 | .05 | .03 | .24 | .08 | .20 | .10 | .46 | .04 | .21 | .01 | **6** |
| **SparseGPT** | .01 | .09 | .14 | .03 | .04 | .01 | .23 | .14 | .48 | .02 | .03 | .07 | 4 |
| **AWQ** | .01 | .07 | .00 | .04 | .01 | .08 | .16 | .11 | .26 | .05 | .00 | .14 | **12** |
| **GPTQ** | .21 | .16 | .42 | .36 | .03 | .37 | .40 | .20 | .59 | .36 | .11 | .27 | 0 |
| *# Wins* | **2** | 1 | 1 | 1 | **2** | 1 | 0 | **4** | 0 | 1 | **3** | 1 | |

| | Strict Edit Locality | | | | | | MMLU | | | | | | |
| | Order-free Error (↓) | | | Order Sensitivity (↓) | | | Order-free Error (↓) | | | Order Sensitivity (↓) | | | |
| Method | FT | MEMIT | LoRA | FT | MEMIT | LoRA | FT | MEMIT | LoRA | FT | MEMIT | LoRA | # Wins |
|---|---|---|---|---|---|---|---|---|---|---|---|---|---|
| **Wanda** | .89 | .97 | .96 | .04 | .00 | .01 | .40 | .41 | .40 | .00 | .00 | .00 | 0 |
| **SparseGPT** | .89 | .96 | .96 | .03 | .00 | .00 | .40 | .40 | .40 | .00 | .00 | .00 | **4** |
| **AWQ** | .85 | .96 | .93 | .04 | .00 | .03 | .41 | .41 | .41 | .02 | .01 | .00 | **5** |
| **GPTQ** | .89 | .96 | .95 | .07 | .00 | .01 | .41 | .41 | .42 | .00 | .01 | .01 | 2 |
| *# Wins* | **4** | 0 | 0 | 0 | **4** | 0 | 0 | 0 | 0 | 1 | 0 | 1 | |

Table 2: Composability metrics for Knowledge Editing composed with Model Compression. Order-free Error (↓) measures best-case error, while Order Sensitivity (↓) measures unwanted effects of intervention re-ordering. # Wins is the number of times each method outperforms others in its category, counting all ties as losses. Sparsity is 25% and quantization is 4-bit. **Key Takeaways:** AWQ composes better than GPTQ and LoRA is surprisingly non-composable.

④ **Overall utility evaluations fail to measure composability.** Despite similar MMLU scores, methods vary significantly in composability. For example, Table 2 shows that MEMIT and Wanda achieve near-perfect Order Sensitivity on MMLU but have 0.24 Order Sensitivity on Edit Success. This is bolstered by Figure 2, where all compositions are similar according to MMLU, but not according to editing metrics. The disparity between overall utility and composability metrics underscores the need for more nuanced evaluations explicitly targeting composability as a design criterion.

### 4.2 COMPOSING MODEL COMPRESSION WITH MACHINE UNLEARNING

⑤ **Compression hinders unlearning.** Our main results in Figure 3 show that applying unlearning to a compressed model can degrade unlearning performance, measured via WMDP (Section 3.1). With RMU, pruning the model before unlearning can significantly regress unlearning performance, especially at higher sparsity levels. There is a similar trend with quantization when composing RMU and GPTQ. With GD, we similarly find that pruning after unlearning is best, though quantizing first can be best. We observe the opposite trend with GA, though GA's poor unlearning performance, as seen in other studies (Maini et al., 2024; Zhang et al., 2024; Wang et al., 2024), is a possible confounder. Overall, unlearning a compressed model is hard, but the degree to which post-compression unlearning hurts performance is contingent on the composition and compression level.

⑥ **Order Sensitivity can determine overall composability.** Table 3 reports the composability metrics of each unlearning technique. Both GD and RMU's successful unlearning (low Order-free Error), with RMU having lower overall Order Sensitivity than GD. While task performance is critical, it alone does not fully capture the practical challenges of applying interventions—RMU is more robust to order.

### 4.3 COMPOSING KNOWLEDGE EDITING WITH MACHINE UNLEARNING

We next examine the composability between editing and unlearning. Both categories of interventions aim to modify the model's knowledge without degrading overall utility. The need to keep LMs factually updated while forgetting undesirable knowledge makes pairing editing and unlearning a practical composition.

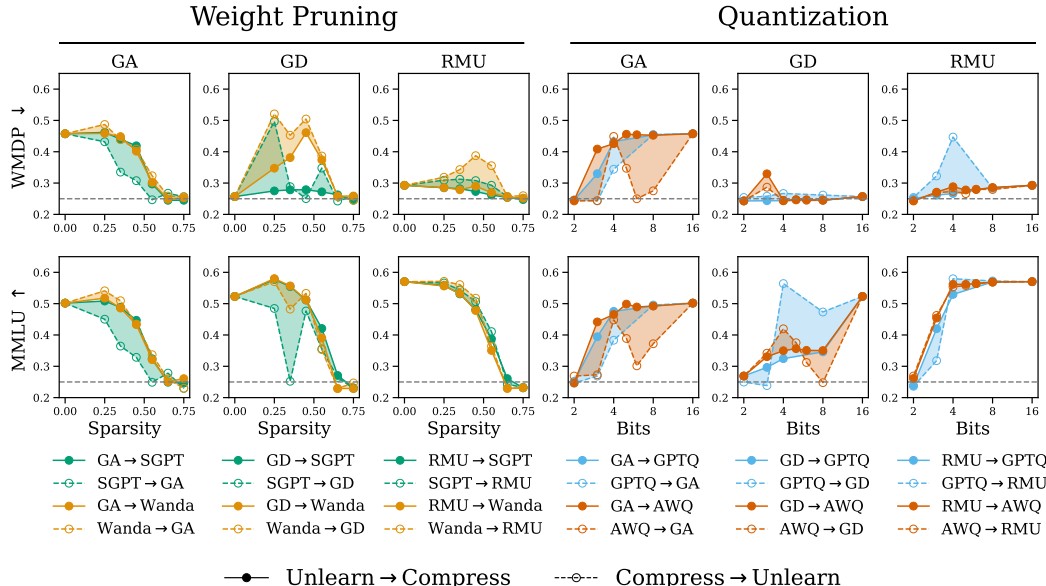

Figure 3: Composing Machine Unlearning with Model Compression at different levels of compression. **Key Takeaways:** Unlearning should generally be applied *before* compression, and performance varies significantly by composition.

| | **WMDP (Unlearning)** | | | | | | **MMLU** | | | | | | |
|---|---|---|---|---|---|---|---|---|---|---|---|---|---|
| | **Order-free Error** (↓) | | | **Order Sensitivity** (↓) | | | **Order-free Error** (↓) | | | **Order Sensitivity** (↓) | | | |
| **Method** | **GA** | **GD** | **RMU** | **GA** | **GD** | **RMU** | **GA** | **GD** | **RMU** | **GA** | **GD** | **RMU** | *# Wins* |
| **Wanda** | .46 | .35 | .29 | .03 | .17 | .03 | .46 | .42 | .43 | .02 | .01 | .01 | 4 |
| **SparseGPT** | .43 | .28 | .28 | .03 | .22 | .02 | .49 | .42 | .43 | .06 | .09 | .01 | 4 |
| **AWQ** | .43 | .24 | .27 | .02 | .00 | .01 | .53 | .58 | .44 | .02 | .07 | .01 | **5** |
| **GPTQ** | .34 | .24 | .27 | .09 | .02 | .18 | .52 | .44 | .42 | .09 | .24 | .05 | 4 |
| *# Wins* | 0 | **2** | 1 | 0 | **2** | 1 | 0 | 2 | 2 | 0 | 0 | **3** | |

Table 3: Composability metrics for machine unlearning and model compression. **Key Takeaways:** GD and RMU have overall comparable performance and far outperform GA. RMU is less sensitive to ordering than GD.

⑦ **Editing and unlearning are highly composable for some unlearning methods.** Table 4 shows that the editing metrics' Order-free Error and Order Sensitivity vary greatly between unlearning techniques. RMU has a lower overall Order-free Error and Order Sensitivity than every other unlearning technique across all editors. Similar to Section 4.2, we find that GD can perform comparably to RMU but with a worse Order Sensitivity. GA's catastrophic forgetting makes it unsuitable for composition. Excluding GA, editing does not disrupt the model's ability to forget the unlearning target, possibly due to knowledge editing's relatively surgical modification to the LM and targeting knowledge semantically distinct from the WMDP unlearning target. Together, these results imply that editing and unlearning can be composable, but composability depends most on the unlearning technique.

## 4.4 TRENDS AND GENERALIZATION ACROSS MODELS

We finally summarize trends revealed when considering all findings.

**Ablation experiments with Mistral 7B Instruct and Yi 1.5 9B Chat show that composition metrics are generally consistent across LMs.** The exception being Editing↔Compression,

| | Edit Success | | | | | | Edit Generalization | | | | | | |
| --- | --- | --- | --- | --- | --- | --- | --- | --- | --- | --- | --- | --- | --- |
| | Order-free Error (↓) | | | Order Sensitivity (↓) | | | Order-free Error (↓) | | | Order Sensitivity (↓) | | | |
| Method | FT | MEMIT | LoRA | FT | MEMIT | LoRA | FT | MEMIT | LoRA | FT | MEMIT | LoRA | # Wins |
| **GA** | .93 | .52 | .00 | .07 | .48 | 1.0 | .96 | .59 | .22 | .04 | .41 | .78 | 1 |
| **GD** | .01 | .07 | .00 | .67 | .40 | .56 | .19 | .11 | .29 | .56 | .41 | .48 | 0 |
| **RMU** | .00 | .03 | .00 | .01 | .01 | .00 | .18 | .07 | .29 | .03 | .04 | .04 | **10** |
| *# Wins* | 0 | 0 | **3** | 1 | 1 | 1 | 0 | **2** | 1 | **2** | 1 | 0 | |

| | WMDP (Unlearning) | | | | | | MMLU | | | | | | |
| --- | --- | --- | --- | --- | --- | --- | --- | --- | --- | --- | --- | --- | --- |
| | Order-free Error (↓) | | | Order Sensitivity (↓) | | | Order-free Error (↓) | | | Order Sensitivity (↓) | | | |
| Method | FT | MEMIT | LoRA | FT | MEMIT | LoRA | FT | MEMIT | LoRA | FT | MEMIT | LoRA | # Wins |
| **GA** | .47 | .40 | .28 | .00 | .05 | .07 | .47 | .51 | .64 | .01 | .04 | .07 | 1 |
| **GD** | .29 | .26 | .30 | .00 | .02 | .24 | .41 | .42 | .41 | .18 | .22 | .14 | 4 |
| **RMU** | .28 | .29 | .29 | .04 | .01 | .00 | .43 | .44 | .44 | .01 | .00 | .04 | **5** |
| *# Wins* | 1 | 1 | 1 | **2** | 0 | 1 | **2** | 0 | 0 | 1 | 1 | 1 | |

Table 4: Composability metrics for Machine Unlearning composed with Model Editing. We study compression at 25% sparsity and 4-bit quantization. Note that lower values of WMDP MCS indicate more unlearning and therefore better performance while higher values are better for the other metrics' MCS. **Key Takeaways:** Editing and unlearning are successfully composable for GD and RMU, with RMU having the lowest Order Sensitivity.

as expected since performance has been shown to vary between models (Meng et al., 2023). Editing↔Unlearning and Compression↔Unlearning consistently exhibit low Order Sensitivity. These results (detailed in Appendix E.1) suggest intervention composability trends often generalize across similar models. We also experiments across model families and sizes in Appendix E.2.

**Compression consistently hinders other interventions.** Compressing an edited model can regress editing performance. Unlearning on a compressed model is often more challenging than on the base model. The same trends appear with triplet interventions, Appendix E.4.1. These trends suggest compression may alter how models encode knowledge, making targeted updates harder. Previous works have explored how compression leads to knowledge loss (Hooker et al., 2019; Du et al., 2021; Azeemi et al., 2023; Hoang et al., 2023) and how editing modifies model internals, increasing fragility (Brown et al., 2023a; Gu et al., 2024b). However, the relationship between *how models are compressed* and *how knowledge is modified* remains underexplored. Understanding these dynamics could guide more composable interventions and illuminate how LMs encode knowledge.

**MMLU is insufficient for measuring composability.** MMLU is a common measure of a model's general performance, but similar scores can mask significant differences in intervention performance. Interventions generally appear more composable on MMLU than on other metrics. Thus, we advocate for thorough multi-metric and multi-dataset evaluations for composable interventions.

## 5 CONCLUSION

We conduct an extensive study of composing test-time interventions for language models. Our setup aligns with the practical need to update the same models multiple times as use cases evolve and training remains expensive. We deploy this new framework to investigate composability between popular interventions spanning knowledge editing, machine unlearning, and model compression. We show that these interventions interact in non-trivial ways, pointing to concrete directions for future work. We further discover guidance for current practice: model compression harms other interventions, unlearning compressed models is especially hard, and the order in which interventions are applied changes their outcomes substantially. Our composability metrics also pave the way for formal investigations and practical principles. Finally, our framework is general and naturally extends to a broad range of test-time interventions.

ACKNOWLEDGEMENTS

We thank the University of Virginia Research Computing team for providing access to excellent high-performance computing resources.

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

APPENDIX

## A  LIMITATIONS

While our work provides the first look at how state-of-the-art methods compose, there are far more possible combinations of interventions, methods, models, and datasets than we can consider in one work. So some popular methods are omitted. We also prioritize broad experiments with many different intervention methods. We hope future works will closely examine individual methods to study, improve, and extend beyond the mechanisms that drive their composability. While we study how composability can vary across similar LMs (Section E.1), it is unknown whether our results generalize across model scale. We hope our codebase empowers others to explore more model sizes and architectures in future research.

## B  BACKGROUND ON INTERVENTIONS

Designing interventions for language models is a highly-active research area. We focus on three key types of intervention.

**Knowledge Editing**. Knowledge editors aim to address factuality decay in LLMs: As the world changes, some facts an LLM learned during training become inaccurate (Mazzia et al., 2023; Wang et al., 2023c). For example, a model trained before 2020 would still predict `The latest pandemic in the US is` → "`Swine Flu`" until updated to predict "`COVID`". There are four general approaches to model editing: 1) Memory-based methods that cache knowledge (Mitchell et al., 2022b; Hartvigsen et al., 2023), 2) Locate-then-Edit methods that selectively finetune parameter subsets (Meng et al., 2022; 2023), 3) Hypernetwork methods that predict new model weights (Sinitsin et al., 2020; Mitchell et al., 2022a; Tan et al., 2024), and 4) Prompt editors that add facts to prompts (Zhong et al., 2023b; Cohen et al., 2024). Most editors update singular facts (Mitchell et al., 2022a; De Cao et al., 2021; Sinitsin et al., 2020; Meng et al., 2022), though recent works apply many simultaneously (Meng et al., 2023; Mitchell et al., 2022b; Tan et al., 2024). Others works embrace the practical need for *sequential* edits (Huang et al., 2023; Hartvigsen et al., 2023). Recent works have also begun investigating unintended impacts of editing on pre-trained models (Gu et al., 2024a; Hoelscher-Obermaier et al., 2023; Li et al., 2023; Hazra et al., 2024; Brown et al., 2023b; Hase et al., 2023), while others are exploring multi-hop editing (Powell et al., 2024; Zhong et al., 2023b; Cohen et al., 2024). However, these works study impacts on model performance, not interactions with other types of interventions.

**Machine Unlearning.** Machine Unlearning (MU) refers to a broad set of techniques for modifying a model post-training to remove the influence of specific training examples or other undesirable properties[4] (Cao & Yang, 2015; Shaik et al., 2023; Liu et al., 2024a). These undesirable properties are commonly referred to as the unlearning target. Crucially, MU must excel at causing the model to forget the unlearning target without adversely affecting overall utility by reducing performance on other domains (Gu et al., 2024c). Another requirement is that MU must cause models to deeply forget knowledge such that the unlearning target can not be elicited by adversaries or probing (Lynch et al., 2024). With the rise of generative models, recent work has explored MU for forgetting potentially copyrighted work (Eldan & Russinovich, 2023), information about individuals (Maini et al., 2024), and potentially dangerous knowledge (Zhao et al., 2023a; Liu et al., 2024b; Li et al., 2024a). Existing works have studied which types of examples models forget when compressed (Hooker et al., 2019), whether compression hinders unlearning (Jia et al., 2023), and whether compression in itself can be used as an unlearning technique (Jia et al., 2023). While related to knowledge editing, the two areas have largely developed independently.

---

[4]Most MU techniques for LMs are considered approximate unlearning in that they aim to modify the model to approximate the behavior of it not being trained on the unlearning target without retraining from scratch.

**Model Compression**. To mitigate the huge computational and memory demands of LLMs, plenty of classical techniques in model compression have been explored, such as quantization (Frantar et al., 2023; Lin et al., 2023; Xiao et al., 2023), network pruning (Frantar & Alistarh, 2023; Sun et al., 2023; Yin et al., 2023; Ma et al., 2023), knowledge distillation (Wang et al., 2023a; Saha et al., 2023; Hsieh et al., 2023), and low-rank factorization (**?**Xu et al., 2023a; Sharma et al., 2023). We focus on quantization and network pruning, which both make in-place updates to pre-trained model weights. Quantization involves lowering the bit-precision of model parameters, effectively shrinking model size and expediting inference. Early works, such as ZeroQuant (Yao et al., 2022) and LLM.int8 (Dettmers et al., 2022), focused on fine-tuning quantization granularity, showing promising results predominantly at higher bit-widths, like 8-bit. More recent innovations like GPTQ (Frantar et al., 2023) and AWQ (Lin et al., 2023) have successfully pushed the boundaries by reducing bit-width to as low as 3 or 4 bits per weight, with negligible performance degradation. Pruning is another compression methods, which aims to eliminate redundant components like weights and channels. Techniques like SparseGPT (Frantar & Alistarh, 2023) and Wanda (Sun et al., 2023) successfully maintain much of an LLM's performance, even when discarding approximately 50% of its weights. OWL (Yin et al., 2023) achieves higher levels of sparsity by reallocating sparsities across layers.

## C    IMPLEMENTATION DETAILS

In this section we outline the set up and hyperparameters used for knowledge editing and compression: pruning and quantization.

### C.1    KNOWLEDGE EDITING DETAILS

#### C.1.1    KNOWLEDGE EDITING METRICS

We adopt three widely-used editing metrics and three compression metrics to measure the effectiveness of these interventions. The editing metrics are:

• *Edit Success:* Assesses the post-edit model's ability to provide the correct, expected answer. An issue arises here with the different levels of verbosity for the models. An unedited model might return the correct token, but use a differently phrased answer or arrange the correct tokens differently. We therefore adopt and F1-style measure used in recent work (Hartvigsen et al., 2023) for this.

First, we determine the common tokens between the generated response and the ground truth, which are represented by the sets $G$ (generated response tokens without special tokens) and $T$ (target tokens without padding), respectively. The common tokens are given by $C = G \cap T$.

If $C$ is empty is empty, the F1 score is set to 0, as there are no common elements to compare.

Otherwise, we calculate precision and recall. Precision is the ratio of the number of common tokens to the number of tokens in the generated response, and recall is the ratio of the number of common tokens to the number of target tokens. They are defined as:

$$\text{Precision} = \frac{|C|}{|G|},$$

$$\text{Recall} = \frac{|C|}{|T|}.$$

The F1 score, which is the harmonic mean of precision and recall, is then calculated as:

$$F1 = \frac{2 \times \text{Precision} \times \text{Recall}}{\text{Precision} + \text{Recall}},$$

$$= \frac{2 \times |C|/|G| \times |C|/|T|}{|C|/|G| + |C|/|T|},$$

with the condition that if the denominator (Precision + Recall) is zero, the F1 score is set to 0.

The final F1 score is the average of the F1 scores for all items. If there are no F1 scores (i.e., the list of scores is empty), the average F1 score is set to 0.

• *Edit Generalization:* Measures the model's capability to genuinely update knowledge, rather than just repeating keywords. This is tested by evaluating edit success on paraphrased prompts given by the respective dataset.

• *Edit Locality:* Examines the precision of edits, ensuring that only relevant, in-distribution knowledge is altered while out-of-distribution facts remain unaffected. In this setting, the F1 score is critical as testing on the pre-trained knowledge can result in varied answer phrasings.

### C.1.2 KNOWLEDGE EDITING DATASET

• *ZsRE* (Levy et al., 2017): This dataset focuses on context-free question-answering. Given a question related to a subject and a relation, the model's task is to provide the correct object as the answer. (Wang et al., 2023b) follow the procedure outlined in (Yao et al., 2023), to improve the locality sets used. As the initial locality sets were solely based on Natural Question annotations, we use the locality sets provided by (Wang et al., 2023b).

In all three datasets, original prompt rephrasings and locality sets are used to derive the respective metrics.

### C.2 MODEL EDITING DETAILS

### C.2.1 MEMIT DETAILS

We use the state-of-the-art MEMIT (Meng et al., 2023) model editor, which applies batches of edits simultaneously. The editing process was applied to layers 4 through 8 of the model, with a clamp normalization factor set at 4. The learning parameters adhered closely to the original implementation: v_num_grad_steps was set to 25, accompanied by a learning rate (lr) of 0.5, and using the last layer for loss calculation. Additionally, a weight decay (weight_decay) of 0.001 was employed. The KL divergence contribution to the overall loss was controlled by a $KL\_$factor of 0.0625. Moreover, a second momentum adjustment was enabled, with an update weight of $15000$, to fine-tune the optimization process. The model generated a maximum length of 40 tokens and a batch size of 50, matching the number of edits being made. 10 repeats were made for each edit and the results averaged.

### C.2.2 LoRA DETAILS

We also employed LoRA as a model editor, specifically using the adalora variant for our experiments. The editing process utilized 70 gradient steps, with a learning rate of 0.005. The rank for the low-rank adaptation was set at 8, with a scaling factor of 32. There was no dropout applied to the adaptation matrices, and the normalization constraint was disabled. The target modules for the edits consisted of query and value projection layers. Consistent with the MEMIT method, the model generated sequences with a maximum length of 40 tokens and utilized a batch size of 50, matching the number of edits per batch. Each edit was repeated 10 times, and the results were averaged to ensure reliability. The data type for the computations was set to `torch.float`.

### C.2.3 FINE-TUNING (FT) DETAILS

For fine-tuning (FT), we focused on layers 4 through 8 of the model. The specific modules targeted for editing included the MLP down-projection (`model.layers.{}.mlp.down_proj`), the general layer module (`model.layers.{}`), the MLP module (`model.layers.{}.mlp`), the self-attention module (`model.layers.{}.self_attn`), the final layer normalization (`model.norm`), and the language model head (`lm_head`).

The fine-tuning process was conducted with 25 optimization steps, using a learning rate of $10^{-5}$. No weight decay was applied. The normalization constraint was disabled. The optimization objective was set to target the new data and the data type for the computations was set to bfloat16.

As with the MEMIT and LoRA methods, the model generated sequences with a maximum length of 40 tokens and used a batch size of 50, matching the number of edits per batch. Each edit was repeated 10 times, and the results were averaged for consistency and reliability.

### C.3 MODEL COMPRESSION DETAILS

### C.3.1 COMPRESSION METRICS

The compression metrics include:

• *Sparsity Ratio.* The fraction of parameters in the model that are zero, indicating the extent to which the model is sparse.

• *Average Bits.* The mean number of bits utilized to represent each parameter in a quantized or sparsified neural network, reflecting the level of compression.

### C.3.2 COMPRESSOR DETAILS

We use four state-of-the-art compression methods including two pruning methods:

• *SparseGPT* (Frantar & Alistarh, 2023): an efficient one-shot pruning method tailored for large models. It converts the pruning process into solving large-scale sparse regression problems using an approximate solver. This approach enables rapid pruning on a single GPU with minimal accuracy loss, achieving 50-60% on large models.

• *Wanda* (Sun et al., 2023): is another popular method for pruning large language models that relies on a pruning metric that combines a weight's magnitude with the norm of its corresponding input activations, determined from a small calibration dataset. The method focuses on selectively pruning weights within individual outputs of linear layers, aiming for high sparsity levels without modifying unpruned weights. Wanda is computationally efficient, executable in a single forward pass.

and two quantization methods:

• *GPTQ* (Frantar et al., 2023): an algorithm designed for efficient weight quantization in large-scale models. It revises the weight quantization approach by quantizing weights in a fixed order rather than a greedy order, which shows minimal performance difference, especially in larger models. GPTQ introduced a novel method where each weight is quantized column-by-column, reducing computational complexity.

• *AWQ* (Lin et al., 2023): is based on the premise that not all weights are equally critical for model performance, and it identifies a small fraction of salient weights whose quantization significantly impacts model accuracy. This identification is done by analyzing activation distributions rather than weight distributions, under the rationale that weights linked to larger activation magnitudes are more crucial.

### C.4 MACHINE UNLEARNING DETAILS

We implement three unlearning techniques: RMU, GA, and GD. The success of each technique hinges on the careful selection of hyperparameters. We describe our parameter searches and selected values in the following sections.

### C.4.1 REPRESENTATION MISDIRECTION UNLEARNING

Introduced in (Li et al., 2024a), RMU intervenes on a specific layer of the LM to perturb activations for inputs related to the unlearning target. This technique has achieved impressive performance

on the WMDP dataset, with performance on the unlearning targets dropping to near-random while overall model utility (MMLU) minimally regresses. Wikitext (Merity et al., 2016) acts as the retain set.

RMU relies on multiple hyperparameters for selecting which layer to modify and how the loss should be calculated. We found that RMU is quite sensitive to hyperparameter choice, with most combinations either leaving the model unaffected or significantly harming model utility. Table 5 details the results from our hyperparameters search for Llama-3 8B. We otherwise use the RMU repo's[5] default settings.

| Parameter | Description | Search | Selected |
|---|---|---|---|
| Alpha | The weight for the retain loss | [1, 10, 100, 1000, 10000] | 1000 |
| Layer | The layer to modify | [3, 17] | 3 |
| Maximum Batches | Training duration RMU in batches | [100, 150, 200, 250, 300] | 250 |

Table 5: **RMU Hyperparameter Search.** Details for the grid search over possible RMU settings. Hyperparameters were selected using grid search.

### C.4.2 GRADIENT ASCENT

GA, a simple and widely studied unlearning method (Golatkar et al., 2019; Wang et al., 2024; Maini et al., 2024; Yao et al., 2024; Zhang et al., 2024), compels the model to minimize the likelihood of the correct token. The underlying intuition is that models will learn to avoid generating tokens associated with the unlearning target. We only train the final 16 layers of the model due to memory constraints.

| Parameter | Description | Search | Selected |
|---|---|---|---|
| LR | The learning rate | [5e-6, 1e-6, ... 5e-3, 1e-3] | 5e-5 |
| Training Samples | Train size balanced across Cyber/Bio | [10, 25, 50, 100, ... 500, 1000, 2000] | 50 |

Table 6: **GA Hyperparameter Search.** Details for the grid search over possible GA settings. Hyperparameters were selected using grid search.

### C.4.3 GRADIENT DIFFERENCE

A significant challenge when leveraging GA is balancing reducing performance on the unlearning target while retaining overall model utility. Since GA reduces the likelihood of the correct token, it is fragile to catastrophic forgetting, where the model incorrectly generalizes to forgetting how to model natural language.

GD (Maini et al., 2024) aims to address this shortcoming by adding a retain loss term to the loss function. This term aims to maximize the likelihood of the correct token on an unrelated retain dataset. We use Wikitext (Merity et al., 2016) as the retain set. As with GA, we only train the final 16 layers of the model.

Table 7 reports our hyperparameter search. Unlike (Maini et al., 2024), which found that GD provided only a marginal improvement over GA, we found that GD can perform significantly better if we upweight the retain loss term. Upweighting is necessary since it is easy for the model to significantly minimize the correct token likelihood, which causes it to drown out the retain loss term. Assigning a weight of 400 to the retain term solved this issue. This hyperparameter combination can make GD competitive with RMU, though still less composable (Section 4).

| Parameter | Description | Search | Selected |
|-----------|-------------|--------|----------|
| Alpha | The weight for the retain loss | [1, 2, 4, 6, 8, 10, 20, 40, 60, 80, 100] | 40 |
| LR | The learning rate | [5e-6, 1e-6, ... 5e-3, 1e-3] | 5e-5 |
| Training Samples | Train size balanced across Cyber/Bio | [10, 25, 50, 100, ... 500, 1000, 2000] | 50 |

Table 7: **GD Hyperparameter Search.** Details for the grid search over possible GD settings. Hyperparameters were selected using grid search.

| | Edit Success | | Edit Generalization | | WMDP | | MMLU | |
|---|---|---|---|---|---|---|---|---|
| | Order-free Error ($\downarrow$) | Order Sensitivity ($\downarrow$) | Order-free Error ($\downarrow$) | Order Sensitivity ($\downarrow$) | Order-free Error ($\downarrow$) | Order Sensitivity ($\downarrow$) | Order-free Error ($\downarrow$) | Order Sensitivity ($\downarrow$) |
| **Different Layers** | .03 | .01 | .07 | .04 | .29 | .01 | .44 | .00 |
| **Overlapping** | .02 | .00 | .06 | .01 | .27 | .00 | .46 | .00 |

Table 8: **Composability Layer Ablation.** We evaluate how layer overlap affects the composability of RMU and MEMIT interventions. Our results demonstrate that composability metrics remain consistent regardless of whether the modified layers overlap. This finding suggests that intervention design, rather than the absence of layer interference, is the primary factor determining the composability of these interventions.

## D   Does Layer Selection Explain Composability?

We examine whether intervention composability stems from targeted modifications to distinct model components or from intrinsic algorithmic design. Table 4 shows that RMU and MEMIT achieve substantially lower Order Sensitivity compared to other intervention pairings. In our primary setup, RMU modifies Llama's 3rd layer, while MEMIT targets layers 4 through 8, raising questions about whether their composability depends on this layer separation.

To isolate this effect, we conducted experiments where both RMU and MEMIT modify the same model layers. We report our results in Table 8 Our findings reveal that these interventions maintain high composability even with overlapping layer targets. The slight increase in Order Sensitivity under layer overlap remains negligible compared to the substantial sensitivity differences observed between other compositions in Section 4.3.

Our results demonstrate that intervention composability stems primarily from algorithmic precision rather than layer isolation. Both RMU and MEMIT achieve effectiveness through targeted modifications: RMU by selectively redirecting activations only for unlearning-targeted inputs, and MEMIT by precisely modifying parameters mediating specific factual associations. This explains composability when each technique modifies semantically distinct knowledge. This investigation reveals that successful composable interventions depend more on specificity and surgical precision than on segregating modifications to separate model components. Effective interventions should precisely target relevant behaviors while minimizing collateral effects on unrelated capabilities.

## E   Extended Results

### E.1   Composability Across Language Models

We investigate the generalizability of intervention composability across different LMs by extending our study from Llama-3 8B to Yi 1.5 9B Chat (Young et al., 2024) and Mistral 7B Instruct (Jiang et al., 2023). These popular open-weights LMs were chosen for their similarity in size and performance to Llama-3. We sample one intervention from each category: MEMIT (editing), AWQ (quantization), Wanda (pruning), and RMU (unlearning). We are unable to find a set of effective hyperpameters for RMU with Yi.

---

[5]https://github.com/centerforaisafety/wmdp

| Composition | Edit Success | | | Edit Generalization | | | Unlearn WMDP | | | MMLU | | | |
|---|---|---|---|---|---|---|---|---|---|---|---|---|---|
| | Llama | Mistral | Yi | Llama | Mistral | Yi | Llama | Mistral | Yi | Llama | Mistral | Yi | *Std Dev* |
| MEMIT↔Wanda | .24 | .00 | .01 | .21 | .02 | .01 | .01 | .01 | .01 | .00 | .01 | .00 | .08 |
| MEMIT↔AWQ | .01 | .11 | .01 | .00 | .04 | .01 | .00 | .01 | .01 | .01 | .01 | .01 | .03 |
| MEMIT↔RMU | .01 | .03 | - | .04 | .02 | - | .01 | .00 | - | .00 | .01 | - | .01 |
| RMU↔AWQ | .01 | .00 | - | .00 | .00 | - | .02 | .04 | - | .01 | .00 | - | .01 |
| RMU↔Wanda | .00 | .00 | - | .01 | .00 | - | .03 | .00 | - | .01 | .00 | - | .01 |

Table 9: Order Sensitivity across LMs. We study compression at 25% sparsity and 4-bit quantization. **Key Takeaway:** Order Sensitivity is generally consistent across LMs, with the exception of compositions involving editing and compression techniques (e.g., MEMIT↔Wanda).

Table 9 shows Order Sensitivity is generally consistent across LMs, except for Editing↔Compression. Editing↔Unlearning and Compression↔Unlearning consistently exhibit low Order Sensitivity. These results suggest intervention composability trends often generalize across similar models, but the choice of LM can remain a hypepramateer for interventions such as Editing.

Figure 4 presents a comprehensive comparison of composition metrics across the three models. The performance of each model is visualized using a bar chart, where the bottom of each bar represents the Order-free Error metric and the height of the bar indicates the Order Sensitivity metric. This visualization allows for a clear assessment of both individual metrics and overall model performance. Notably, the results demonstrate that composability generalizes across different models This finding underscores the robustness of composability as a general property while highlighting the nuanced impact of specific model architectures on edit generalization capabilities.

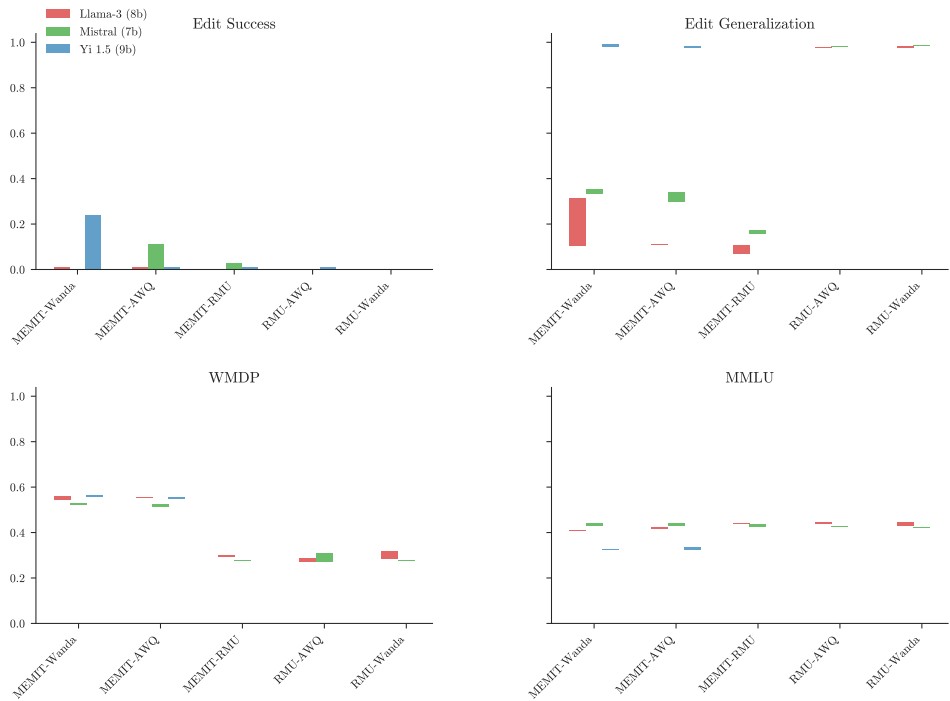

Figure 4: Comparison of Composition Metrics across different models. The performance of each model is represented by a bar. The bottom of the bar represents the Order-free Error and the size of the bar represents the Order Sensitivity. **Key Takeaways:** Composability generalizes overall across models, with Edit Generalization having an expected model-dependent effect.

## E.2 Additional Experiments Across Model Families and Sizes

We extend our analysis to examine composability across different model variants to understand how these properties generalize beyond our main experiments. Specifically, we investigate two dimensions: (1) comparing base vs. instruction-tuned variants of the same model, and (2) examining how parameter count affects composability.

### E.2.1 Base vs. Instruction-Tuned Models

To investigate whether our findings generalize across training regimes, we compare the base Llama-3 8B with its instruction-tuned counterpart (Llama-3 8B Instruct). We select representative interventions from each category: MEMIT (editing), WANDA (compression), and RMU (unlearning).

Table 10 shows key composability metrics across these model variants. We find that our main composability findings remain consistent between base and instruction-tuned models, suggesting that architecture and pretraining regime influence composability more than instruction tuning.

Table 10: Performance comparison between base and instruction-tuned models. Both models exhibit similar composability patterns, though with different absolute performance.

| Llama-3 8B version | Intervention Sequence | MMLU | Avg. WMDP | Edit Success | Edit Generalization | Edit Locality |
|---|---|---|---|---|---|---|
| Base | MEMIT → WANDA | 58.89% | – | 70.11% | 68.81% | 3.05% |
| Instruct | MEMIT → WANDA | 61.12% | – | 93.93% | 84.47% | 2.69% |
| Base | WANDA → MEMIT | 59.22% | – | 94.59% | 89.61% | 3.44% |
| Instruct | WANDA → MEMIT | 61.47% | – | 99.10% | 84.77% | 3.26% |
| Base | RMU → WANDA | 55.64% | 28.54% | – | – | – |
| Instruct | RMU → WANDA | 55.66% | 26.82% | – | – | – |
| Base | WANDA → RMU | 57.12% | 31.86% | – | – | – |
| Instruct | WANDA → RMU | 58.21% | 31.83% | – | – | – |
| Base | MEMIT → RMU | 56.19% | 29.28% | 95.62% | 89.38% | 3.48% |
| Instruct | MEMIT → RMU | 54.59% | 27.28% | 93.88% | 87.00% | 3.19% |
| Base | RMU → MEMIT | 55.94% | 30.14% | 96.87% | 92.97% | 3.40% |
| Instruct | RMU → MEMIT | 53.99% | 25.53% | 89.41% | 84.44% | 3.43% |

Three key findings remain consistent:

**Compression → Editing is Best:** For both base and instruct models, applying WANDA before MEMIT leads to significantly better edit success than the reverse order. Interestingly, the instruct model shows more robustness to ordering, with edit success for MEMIT→WANDA at 93.93% compared to 70.11% for the base model.

**Unlearning → Compression is Best:** Both model variants demonstrate better WMDP scores (lower is better) when applying RMU before WANDA rather than the reverse order. The performance difference remains similar across variants: approximately 3.3% for base and 5.0% for instruct.

**Editing↔Unlearning is Composable:** MEMIT and RMU remain largely composable in both model variants, though the instruct model shows a slightly larger performance delta between order permutations on the editing metrics.

### E.2.2 Effect of Model Size

We further investigate whether our findings generalize across model scales by comparing Llama-3 8B with the smaller Llama-3.2 1B. Table 11 shows a comparison of performance for editing and compression interventions across these model scales.

We find that the smaller 1B Llama shows similar composability trends to its larger 8B counterpart, with an average order sensitivity of 3.60% compared to 2.50% for the 8B model. The preference for applying editing after compression remains consistent across model sizes, though the effect is more pronounced in the 8B model. While absolute MMLU performance is understandably lower

Table 11: Comparison of performance across different Llama model sizes (8B vs 1B). Despite significant differences in parameter count, key composability trends remain consistent.

| Size | Intervention Sequence | MMLU | Edit Success | Edit Generalization |
|------|----------------------|------|--------------|---------------------|
| 8B | LoRA → WANDA | 60.00% | 87.00% | 53.00% |
| 1B | LoRA → WANDA | 34.03% | 98.00% | 64.96% |
| 8B | FT → WANDA | 60.00% | 96.00% | 76.00% |
| 1B | FT → WANDA | 35.01% | 90.26% | 75.06% |
| 8B | WANDA → LoRA | 60.00% | 95.00% | 54.00% |
| 1B | WANDA → LoRA | 34.03% | 100.00% | 76.76% |
| 8B | WANDA → FT | 60.00% | 99.00% | 80.00% |
| 1B | WANDA → FT | 33.82% | 100.00% | 65.26% |

for the 1B model, editing metrics remain competitive, suggesting that composability properties may transfer across model scales within the same architecture family.

### E.3 STATISTICAL ANALYSIS OF RESULTS

We supplement our composability analysis with statistical measures to quantify the reliability of our findings. Our editing results represent averages over 10 random data subsets, while compression methods are deterministic. For unlearning, we observed minimal variance in replicated experiments, with standard deviations of approximately $\pm0.5\%$ for accuracy metrics.

Table 12 presents our composability metrics with standard errors for knowledge editing composed with model compression. The low standard errors across metrics indicate robust findings that are not overly sensitive to specific data subsets.

Table 12: Composability metrics with standard errors (in parentheses) for Knowledge Editing composed with Model Compression. Order-free Error ($\downarrow$) measures best-case error caused by interventions, while Order Sensitivity ($\downarrow$) measures unwanted effects of intervention ordering.

| | Edit Success | | | | | | Edit Generalization | | | | | | |
|--------|-----------|--------|-------|-----------|--------|-------|-----------|--------|-------|-----------|--------|-------|--------|
| | Order-free Error ($\downarrow$) | | | Order Sensitivity ($\downarrow$) | | | Order-free Error ($\downarrow$) | | | Order Sensitivity ($\downarrow$) | | | |
| Method | FT | MEMIT | LoRA | FT | MEMIT | LoRA | FT | MEMIT | LoRA | FT | MEMIT | LoRA | # Wins |
| Wanda | .01 (.01) | .05 (.02) | .05 (.01) | .03 (.01) | .24 (.03) | .08 (.02) | .20 (.02) | .10 (.01) | .46 (.03) | .04 (.01) | .21 (.03) | .01 (.01) | 6 |
| SparseGPT | .01 (.01) | .09 (.02) | .14 (.02) | .03 (.01) | .04 (.02) | .01 (.01) | .23 (.02) | .14 (.02) | .48 (.02) | .02 (.01) | .03 (.02) | .07 (.02) | 4 |
| AWQ | .01 (.01) | .07 (.01) | .00 (.01) | .04 (.01) | .01 (.01) | .08 (.01) | .16 (.02) | .11 (.02) | .26 (.03) | .05 (.01) | .00 (.01) | .14 (.02) | 12 |
| GPTQ | .21 (.02) | .16 (.03) | .42 (.03) | .36 (.04) | .03 (.03) | .37 (.03) | .40 (.04) | .20 (.03) | .59 (.04) | .36 (.03) | .11 (.02) | .27 (.03) | 0 |

Standard errors are particularly small for the MMLU metrics ($\leq0.01$), indicating high stability in general utility evaluations. Edit Success and Generalization metrics show slightly higher variance (up to 0.04), as expected given the inherent variability in editing processes. Importantly, the key findings remain statistically reliable despite this variation. For example, the difference in Order Sensitivity between MEMIT and LoRA when paired with Wanda (0.24 vs. 0.08) is substantially larger than their respective standard errors (0.03 and 0.02).

The statistical analysis reinforces our confidence in the identified composability trends and indicates that the observed patterns represent genuine differences in intervention behaviors rather than artifacts of specific data samples.

### E.4 COMPOSING MORE THAN TWO INTERVENTIONS

Our composability framework naturally extends beyond pairs of interventions. While exhaustively exploring all possible combinations of three or more interventions is computationally prohibitive with our resources, we conduct targeted experiments to understand how performance changes when composing triplets of interventions.

For these experiments, we select the most composable intervention from each category based on our earlier findings: WANDA with 25% sparsity (compression), MEMIT (editing), and RMU (unlearning). We examine all six possible orderings of these three interventions, evaluating both task-specific metrics and general model utility.

### E.4.1 EFFECT OF TRIPLET COMPOSITIONS ON PERFORMANCE

A central question is whether composing more than two interventions leads to catastrophic regression in overall performance. Table 13 and Table 14 show the results for both base and instruction-tuned Llama-3 8B models across all possible orderings of our three selected interventions.

Table 13: Performance of triplet intervention compositions on Llama-3 8B Base. Each row represents a different ordering of the three interventions, with interventions applied from left to right.

| Intervention Sequence | MMLU | Avg WMDP | Edit Success | Edit Generalization |
|---|---|---|---|---|
| WANDA → RMU → MEMIT | 56.36% | 30.77% | 99.20% | 93.70% |
| MEMIT → WANDA → RMU | 54.63% | 27.60% | 99.50% | 90.30% |
| MEMIT → RMU → WANDA | 54.17% | 28.00% | 99.50% | 90.12% |
| WANDA → MEMIT → RMU | 55.23% | 31.30% | 95.86% | 90.26% |
| RMU → WANDA → MEMIT | 53.87% | 28.20% | 93.00% | 92.00% |
| RMU → MEMIT → WANDA | 54.32% | 28.20% | 94.67% | 85.17% |

Table 14: Performance of triplet intervention compositions on Llama-3 8B Instruct. Comparison with base model results shows consistent ordering patterns but with different absolute values.

| Intervention Sequence | MMLU | Avg WMDP | Edit Success | Edit Generalization |
|---|---|---|---|---|
| WANDA → MEMIT → RMU | 59.09% | 34.68% | 99.10% | 84.77% |
| MEMIT → WANDA → RMU | 54.27% | 26.98% | 94.60% | 85.27% |
| MEMIT → RMU → WANDA | 54.61% | 27.39% | 91.53% | 80.47% |
| WANDA → RMU → MEMIT | 57.57% | 31.46% | 86.72% | 83.50% |
| RMU → WANDA → MEMIT | 53.77% | 26.41% | 86.00% | 80.77% |
| RMU → MEMIT → WANDA | 54.53% | 26.33% | 74.82% | 81.44% |

Comparing the results across all ordering permutations, we observe several key findings:

**Overall Utility Remains Stable:** MMLU performance shows relatively small variance across different orderings (range of 2.49% for base and 5.32% for instruct), confirming that composing three interventions does not catastrophically degrade general model utility.

**Task-Specific Metrics Show Order Sensitivity:** While general utility remains stable, task-specific metrics like Edit Success show much larger variance (range of 6.50% for base and 24.28% for instruct), highlighting the importance of ordering even in triplet compositions.

**Best Overall Performance:** For the base model, the WANDA → RMU → MEMIT ordering achieves the best balance of metrics, with high values for both edit metrics and reasonable WMDP performance. This is consistent with our findings from pairwise compositions, where compression applied before unlearning and editing typically yields better results.

**Consistency Across Model Variants:** The instruction-tuned model shows patterns similar to the base model, with similar relative ordering preferences, though with different absolute performance values. This further supports our earlier finding that composability patterns tend to be preserved across model variants, even when expanded to triplet interventions.

### E.4.2 IMPLICATIONS FOR PRACTICAL DEPLOYMENT

These results have important implications for deploying multiple interventions in practice. First, they demonstrate that multiple successive interventions can be applied without catastrophic degradation of general model utility. Second, they confirm that the order in which interventions are applied has significant impact on task-specific performance, with some orderings preserving task-specific metrics much better than others.

For practitioners seeking to apply multiple interventions, our results suggest that: (1) compression should typically be applied early in a sequence of interventions, (2) editing should generally be applied last to maximize retention of edited information, and (3) unlearning is most effective when applied between compression and editing.

### E.5 COMPLETE PAIRWISE RESULTS

We report the task performance for all compositions below and the baseline results, where only a single intervention is applied (Table 15). Perplexity on Wikitext (Merity et al., 2016) is also included as another overall utility metric. Average Bits (Appendix C.3.1) measures the degree to which the composition compresses the model.

| | Editing | | | Compression | Unlearning | Utility | |
|---|---|---|---|---|---|---|---|
| | Edit Success | Generalization | Locality | Avg. Bits | Avg. WMDP | MMLU | WikiText PPL |
| None | 0.02 | 0.02 | 0.04 | 16.00 | 0.58 | 0.62 | 5.54 |
| FT→None | 0.99 | 0.82 | 0.11 | 16.0 | 0.57 | 0.61 | 5.57 |
| MEMIT→None | 0.89 | 0.85 | 0.04 | 16.0 | 0.57 | 0.61 | 5.57 |
| LoRA→None | 1.0 | 0.71 | 0.06 | 16.0 | 0.56 | 0.61 | 19.25 |
| SparseGPT (0.25) →None | 0.02 | 0.02 | 0.03 | 12.25 | 0.56 | 0.61 | 5.87 |
| SparseGPT (0.35) →None | 0.02 | 0.02 | 0.03 | 10.75 | 0.54 | 0.58 | 6.34 |
| SparseGPT (0.45) →None | 0.02 | 0.01 | 0.03 | 9.25 | 0.51 | 0.54 | 7.43 |
| SparseGPT (0.55) →None | 0.01 | 0.01 | 0.03 | 7.75 | 0.45 | 0.44 | 10.43 |
| SparseGPT (0.65) →None | 0.0 | 0.01 | 0.03 | 6.25 | 0.3 | 0.27 | 20.83 |
| SparseGPT (0.75) →None | 0.01 | 0.01 | 0.03 | 4.75 | 0.26 | 0.23 | 81.84 |
| Wanda (0.25) →None | 0.02 | 0.02 | 0.03 | 12.25 | 0.56 | 0.61 | 5.84 |
| Wanda (0.35) →None | 0.02 | 0.02 | 0.04 | 10.75 | 0.54 | 0.58 | 6.31 |
| Wanda (0.45) →None | 0.02 | 0.02 | 0.03 | 9.25 | 0.5 | 0.52 | 7.52 |
| Wanda (0.55) →None | 0.02 | 0.01 | 0.02 | 7.75 | 0.36 | 0.37 | 12.5 |
| Wanda (0.65) →None | 0.02 | 0.01 | 0.02 | 6.25 | 0.26 | 0.23 | 47.18 |
| Wanda (0.75) →None | 0.02 | 0.03 | 0.03 | 4.75 | 0.26 | 0.23 | 257.46 |
| GPTQ (2-Bit) →None | 0.0 | 0.0 | 0.03 | 2.25 | 0.25 | 0.24 | 3681.23 |
| GPTQ (3-Bit) →None | 0.01 | 0.01 | 0.03 | 3.25 | 0.45 | 0.46 | 8.6 |
| GPTQ (4-Bit) →None | 0.02 | 0.02 | 0.03 | 4.25 | 0.56 | 0.6 | 9.97 |
| GPTQ (8-Bit) →None | 0.02 | 0.02 | 0.04 | 8.25 | 0.58 | 0.62 | 5.54 |
| AWQ (2-Bit) →None | 0.0 | 0.0 | 0.0 | 2.25 | 0.24 | 0.27 | 1748954.75 |
| AWQ (3-Bit) →None | 0.01 | 0.01 | 0.03 | 3.25 | 0.52 | 0.53 | 7.47 |
| AWQ (4-Bit) →None | 0.02 | 0.02 | 0.03 | 4.25 | 0.57 | 0.6 | 5.91 |
| AWQ (5-Bit) →None | 0.02 | 0.03 | 0.03 | 5.25 | 0.57 | 0.62 | 5.62 |
| AWQ (6-Bit) →None | 0.02 | 0.03 | 0.04 | 6.25 | 0.58 | 0.62 | 5.56 |
| AWQ (8-Bit) →None | 0.02 | 0.02 | 0.04 | 8.25 | 0.58 | 0.62 | 5.54 |
| GA→None | 0.0 | 0.0 | 0.0 | 16.0 | 0.46 | 0.5 | inf |
| GD→None | 0.02 | 0.0 | 0.09 | 16.0 | 0.26 | 0.52 | 4.48 |
| RMU→None | 0.02 | 0.02 | 0.03 | 16.0 | 0.29 | 0.57 | 5.56 |

Table 15: Baseline intervention Results without composition.

| | Editing | | | Compression | Unlearning | Utility | |
|---|---|---|---|---|---|---|---|
| | Edit Success | Generalization | Locality | Avg. Bits | Avg. WMDP | MMLU | WikiText PPL |
| FT→SparseGPT (0.25) | 0.96 | 0.76 | 0.11 | 12.25 | 0.55 | 0.6 | 5.9 |
| FT→SparseGPT (0.35) | 0.84 | 0.66 | 0.09 | 10.75 | 0.53 | 0.57 | 6.38 |
| FT→SparseGPT (0.45) | 0.64 | 0.5 | 0.07 | 9.25 | 0.5 | 0.53 | 7.48 |
| FT→SparseGPT (0.55) | 0.33 | 0.26 | 0.04 | 7.75 | 0.44 | 0.44 | 10.35 |
| FT→SparseGPT (0.65) | 0.12 | 0.1 | 0.03 | 6.25 | 0.33 | 0.3 | 21.54 |
| FT→SparseGPT (0.75) | 0.03 | 0.02 | 0.03 | 4.75 | 0.26 | 0.23 | 88.73 |
| FT→Wanda (0.25) | 0.96 | 0.76 | 0.11 | 12.25 | 0.56 | 0.6 | 5.88 |
| FT→Wanda (0.35) | 0.88 | 0.68 | 0.09 | 10.75 | 0.53 | 0.57 | 6.35 |
| FT→Wanda (0.45) | 0.65 | 0.5 | 0.06 | 9.25 | 0.49 | 0.52 | 7.57 |
| FT→Wanda (0.55) | 0.31 | 0.24 | 0.04 | 7.75 | 0.35 | 0.37 | 12.48 |
| FT→Wanda (0.65) | 0.08 | 0.07 | 0.03 | 6.25 | 0.26 | 0.23 | 46.37 |
| FT→Wanda (0.75) | 0.02 | 0.02 | 0.03 | 4.75 | 0.26 | 0.23 | 344.62 |
| FT→GPTQ (2-Bit) | 0.0 | 0.0 | 0.02 | 2.25 | 0.25 | 0.24 | 1995.71 |
| FT→GPTQ (3-Bit) | 0.27 | 0.2 | 0.04 | 3.25 | 0.43 | 0.46 | 9.03 |
| FT→GPTQ (4-Bit) | 0.79 | 0.6 | 0.11 | 4.25 | 0.54 | 0.59 | 12.93 |
| FT→GPTQ (8-Bit) | 0.99 | 0.83 | 0.11 | 8.25 | 0.57 | 0.61 | 5.57 |
| FT→AWQ (2-Bit) | 0.0 | 0.0 | 0.0 | 2.25 | 0.24 | 0.27 | 1739216.25 |
| FT→AWQ (3-Bit) | 0.7 | 0.55 | 0.07 | 3.25 | 0.5 | 0.51 | 7.56 |
| FT→AWQ (4-Bit) | 0.95 | 0.79 | 0.1 | 4.25 | 0.55 | 0.59 | 5.95 |
| FT→AWQ (5-Bit) | 0.99 | 0.84 | 0.11 | 5.25 | 0.56 | 0.61 | 5.66 |
| FT→AWQ (6-Bit) | 0.99 | 0.84 | 0.1 | 6.25 | 0.57 | 0.61 | 5.59 |
| FT→AWQ (8-Bit) | 0.99 | 0.83 | 0.11 | 8.25 | 0.57 | 0.61 | 5.57 |
| MEMIT→SparseGPT (0.25) | 0.87 | 0.83 | 0.04 | 12.25 | 0.56 | 0.6 | 5.89 |
| MEMIT→SparseGPT (0.35) | 0.81 | 0.74 | 0.03 | 10.75 | 0.54 | 0.57 | 6.4 |
| MEMIT→SparseGPT (0.45) | 0.62 | 0.56 | 0.04 | 9.25 | 0.51 | 0.53 | 7.43 |
| MEMIT→SparseGPT (0.55) | 0.28 | 0.2 | 0.03 | 7.75 | 0.43 | 0.43 | 10.37 |
| MEMIT→SparseGPT (0.65) | 0.03 | 0.03 | 0.03 | 6.25 | 0.3 | 0.29 | 21.02 |
| MEMIT→SparseGPT (0.75) | 0.01 | 0.01 | 0.02 | 4.75 | 0.26 | 0.23 | 71.62 |
| MEMIT→Wanda (0.25) | 0.7 | 0.69 | 0.03 | 12.25 | 0.54 | 0.59 | 5.87 |
| MEMIT→Wanda (0.35) | 0.85 | 0.78 | 0.03 | 10.75 | 0.53 | 0.57 | 6.34 |
| MEMIT→Wanda (0.45) | 0.75 | 0.67 | 0.03 | 9.25 | 0.48 | 0.51 | 7.56 |
| MEMIT→Wanda (0.55) | 0.32 | 0.26 | 0.03 | 7.75 | 0.35 | 0.36 | 12.33 |
| MEMIT→Wanda (0.65) | 0.01 | 0.01 | 0.02 | 6.25 | 0.26 | 0.23 | 46.98 |
| MEMIT→Wanda (0.75) | 0.02 | 0.02 | 0.03 | 4.75 | 0.26 | 0.23 | 290.27 |
| MEMIT→GPTQ (2-Bit) | 0.0 | 0.0 | 0.02 | 2.25 | 0.25 | 0.24 | 1663.86 |
| MEMIT→GPTQ (3-Bit) | 0.35 | 0.29 | 0.04 | 3.25 | 0.45 | 0.46 | 8.89 |
| MEMIT→GPTQ (4-Bit) | 0.84 | 0.8 | 0.03 | 4.25 | 0.55 | 0.59 | 13.17 |
| MEMIT→GPTQ (8-Bit) | 0.94 | 0.9 | 0.04 | 8.25 | 0.57 | 0.61 | 5.56 |
| MEMIT→AWQ (2-Bit) | 0.0 | 0.0 | 0.0 | 2.25 | 0.24 | 0.27 | 1738085.5 |
| MEMIT→AWQ (3-Bit) | 0.83 | 0.79 | 0.04 | 3.25 | 0.5 | 0.5 | 7.54 |
| MEMIT→AWQ (4-Bit) | 0.93 | 0.89 | 0.03 | 4.25 | 0.55 | 0.59 | 5.94 |
| MEMIT→AWQ (5-Bit) | 0.92 | 0.89 | 0.03 | 5.25 | 0.57 | 0.61 | 5.65 |
| MEMIT→AWQ (6-Bit) | 0.93 | 0.89 | 0.04 | 6.25 | 0.57 | 0.61 | 5.59 |
| MEMIT→AWQ (8-Bit) | 0.94 | 0.9 | 0.04 | 8.25 | 0.57 | 0.61 | 5.56 |
| LoRA→SparseGPT (0.25) | 0.86 | 0.52 | 0.04 | 12.25 | 0.55 | 0.6 | 13.81 |
| LoRA→SparseGPT (0.35) | 0.62 | 0.37 | 0.05 | 10.75 | 0.52 | 0.57 | 10.6 |
| LoRA→SparseGPT (0.45) | 0.36 | 0.22 | 0.04 | 9.25 | 0.48 | 0.52 | 16.48 |
| LoRA→SparseGPT (0.55) | 0.17 | 0.14 | 0.04 | 7.75 | 0.43 | 0.43 | 23.46 |
| LoRA→SparseGPT (0.65) | 0.07 | 0.06 | 0.03 | 6.25 | 0.32 | 0.3 | 53.89 |
| LoRA→SparseGPT (0.75) | 0.03 | 0.03 | 0.02 | 4.75 | 0.26 | 0.25 | 186.65 |
| LoRA→Wanda (0.25) | 0.87 | 0.53 | 0.04 | 12.25 | 0.55 | 0.6 | 9.23 |
| LoRA→Wanda (0.35) | 0.64 | 0.4 | 0.05 | 10.75 | 0.52 | 0.57 | 11.01 |
| LoRA→Wanda (0.45) | 0.37 | 0.25 | 0.03 | 9.25 | 0.46 | 0.49 | 12.6 |
| LoRA→Wanda (0.55) | 0.18 | 0.14 | 0.03 | 7.75 | 0.36 | 0.36 | 33.59 |
| LoRA→Wanda (0.65) | 0.05 | 0.04 | 0.02 | 6.25 | 0.26 | 0.24 | 194.26 |
| LoRA→Wanda (0.75) | 0.02 | 0.02 | 0.03 | 4.75 | 0.25 | 0.25 | 914.43 |
| LoRA→GPTQ (2-Bit) | 0.0 | 0.0 | 0.01 | 2.25 | 0.25 | 0.24 | 10465.63 |
| LoRA→GPTQ (3-Bit) | 0.08 | 0.06 | 0.02 | 3.25 | 0.4 | 0.42 | 40.02 |
| LoRA→GPTQ (4-Bit) | 0.58 | 0.41 | 0.05 | 4.25 | 0.54 | 0.58 | 63.45 |
| LoRA→GPTQ (8-Bit) | 1.0 | 0.71 | 0.06 | 8.25 | 0.57 | 0.61 | 18.86 |
| LoRA→AWQ (2-Bit) | 0.0 | 0.0 | 0.0 | 2.25 | 0.24 | 0.25 | 1764088.0 |
| LoRA→AWQ (3-Bit) | 0.46 | 0.31 | 0.05 | 3.25 | 0.5 | 0.51 | 24.29 |
| LoRA→AWQ (4-Bit) | 0.91 | 0.6 | 0.04 | 4.25 | 0.55 | 0.59 | 16.37 |
| LoRA→AWQ (5-Bit) | 0.99 | 0.7 | 0.05 | 5.25 | 0.56 | 0.6 | 16.31 |
| LoRA→AWQ (6-Bit) | 1.0 | 0.71 | 0.06 | 6.25 | 0.56 | 0.61 | 15.34 |
| LoRA→AWQ (8-Bit) | 1.0 | 0.72 | 0.06 | 8.25 | 0.57 | 0.61 | 15.25 |

Table 16: Detailed results for editing → compression.

| | Editing | | | Compression | Unlearning | Utility | |
| --- | --- | --- | --- | --- | --- | --- | --- |
| | Edit Success | Generalization | Locality | Avg. Bits | Avg. WMDP | MMLU | WikiText PPL |
| SparseGPT (0.25) →FT | 0.99 | 0.77 | 0.08 | 12.25 | 0.55 | 0.6 | 5.91 |
| SparseGPT (0.35) →FT | 0.98 | 0.75 | 0.06 | 10.75 | 0.54 | 0.58 | 6.37 |
| SparseGPT (0.45) →FT | 0.96 | 0.71 | 0.06 | 9.25 | 0.51 | 0.54 | 7.49 |
| SparseGPT (0.55) →FT | 0.95 | 0.66 | 0.05 | 7.75 | 0.44 | 0.44 | 10.19 |
| SparseGPT (0.65) →FT | 0.89 | 0.54 | 0.04 | 6.25 | 0.33 | 0.3 | 17.67 |
| SparseGPT (0.75) →FT | 0.58 | 0.25 | 0.04 | 4.75 | 0.26 | 0.24 | 98.18 |
| SparseGPT (0.25) →MEMIT | 0.91 | 0.86 | 0.04 | 12.25 | 0.56 | 0.59 | 5.89 |
| SparseGPT (0.35) →MEMIT | 0.89 | 0.8 | 0.03 | 10.75 | 0.54 | 0.57 | 6.38 |
| SparseGPT (0.45) →MEMIT | 0.62 | 0.48 | 0.03 | 9.25 | 0.51 | 0.53 | 7.49 |
| SparseGPT (0.55) →MEMIT | 0.37 | 0.27 | 0.03 | 7.75 | 0.44 | 0.43 | 10.62 |
| SparseGPT (0.65) →MEMIT | 0.18 | 0.13 | 0.03 | 6.25 | 0.32 | 0.29 | 22.83 |
| SparseGPT (0.75) →MEMIT | 0.11 | 0.08 | 0.04 | 4.75 | 0.25 | 0.23 | 394.59 |
| SparseGPT (0.25) →LoRA | 0.85 | 0.45 | 0.04 | 12.25 | 0.54 | 0.6 | 10.69 |
| SparseGPT (0.35) →LoRA | 0.76 | 0.42 | 0.04 | 10.75 | 0.52 | 0.57 | 9.54 |
| SparseGPT (0.45) →LoRA | 0.64 | 0.33 | 0.04 | 9.25 | 0.48 | 0.52 | 10.52 |
| SparseGPT (0.55) →LoRA | 0.55 | 0.29 | 0.03 | 7.75 | 0.43 | 0.43 | 11.5 |
| SparseGPT (0.65) →LoRA | 0.37 | 0.17 | 0.03 | 6.25 | 0.32 | 0.29 | 23.13 |
| SparseGPT (0.75) →LoRA | 0.08 | 0.06 | 0.03 | 4.75 | 0.26 | 0.25 | 89.47 |
| Wanda (0.25) →FT | 0.99 | 0.8 | 0.07 | 12.25 | 0.56 | 0.6 | 5.86 |
| Wanda (0.35) →FT | 0.97 | 0.76 | 0.05 | 10.75 | 0.54 | 0.58 | 6.32 |
| Wanda (0.45) →FT | 0.94 | 0.69 | 0.04 | 9.25 | 0.49 | 0.52 | 7.52 |
| Wanda (0.55) →FT | 0.8 | 0.5 | 0.03 | 7.75 | 0.36 | 0.38 | 11.98 |
| Wanda (0.65) →FT | 0.5 | 0.28 | 0.03 | 6.25 | 0.26 | 0.23 | 41.41 |
| Wanda (0.75) →FT | 0.05 | 0.04 | 0.03 | 4.75 | 0.26 | 0.23 | 249.29 |
| Wanda (0.25) →MEMIT | 0.95 | 0.9 | 0.03 | 12.25 | 0.56 | 0.59 | 5.87 |
| Wanda (0.35) →MEMIT | 0.86 | 0.76 | 0.03 | 10.75 | 0.54 | 0.57 | 6.34 |
| Wanda (0.45) →MEMIT | 0.72 | 0.59 | 0.03 | 9.25 | 0.49 | 0.51 | 7.54 |
| Wanda (0.55) →MEMIT | 0.37 | 0.26 | 0.03 | 7.75 | 0.37 | 0.37 | 12.07 |
| Wanda (0.65) →MEMIT | 0.04 | 0.04 | 0.02 | 6.25 | 0.26 | 0.23 | 40.49 |
| Wanda (0.75) →MEMIT | 0.04 | 0.04 | 0.02 | 4.75 | 0.26 | 0.23 | 280.04 |
| Wanda (0.25) →LoRA | 0.95 | 0.54 | 0.03 | 12.25 | 0.55 | 0.6 | 10.65 |
| Wanda (0.35) →LoRA | 0.86 | 0.48 | 0.04 | 10.75 | 0.52 | 0.57 | 7.98 |
| Wanda (0.45) →LoRA | 0.74 | 0.43 | 0.04 | 9.25 | 0.47 | 0.5 | 9.18 |
| Wanda (0.55) →LoRA | 0.68 | 0.37 | 0.03 | 7.75 | 0.36 | 0.37 | 15.01 |
| Wanda (0.65) →LoRA | 0.34 | 0.19 | 0.02 | 6.25 | 0.26 | 0.23 | 57.63 |
| Wanda (0.75) →LoRA | 0.06 | 0.06 | 0.03 | 4.75 | 0.26 | 0.23 | 318.06 |
| GPTQ (2-Bit) →FT | 0.0 | 0.0 | 0.01 | 2.25 | 0.25 | 0.24 | 970816.25 |
| GPTQ (3-Bit) →FT | 0.1 | 0.08 | 0.03 | 3.25 | 0.41 | 0.42 | 9.96 |
| GPTQ (4-Bit) →FT | 0.44 | 0.24 | 0.04 | 4.25 | 0.55 | 0.58 | 6.4 |
| GPTQ (8-Bit) →FT | 0.99 | 0.83 | 0.11 | 8.25 | 0.57 | 0.61 | 5.57 |
| GPTQ (2-Bit) →MEMIT | 0.0 | 0.0 | 0.01 | 2.25 | 0.25 | 0.24 | 739248.62 |
| GPTQ (3-Bit) →MEMIT | 0.12 | 0.1 | 0.03 | 3.25 | 0.41 | 0.42 | 9.81 |
| GPTQ (4-Bit) →MEMIT | 0.81 | 0.69 | 0.04 | 4.25 | 0.55 | 0.58 | 6.38 |
| GPTQ (8-Bit) →MEMIT | 0.95 | 0.91 | 0.03 | 8.25 | 0.57 | 0.61 | 5.57 |
| GPTQ (2-Bit) →LoRA | 0.0 | 0.0 | 0.01 | 2.25 | 0.25 | 0.24 | 695958.19 |
| GPTQ (3-Bit) →LoRA | 0.01 | 0.02 | 0.02 | 3.25 | 0.37 | 0.38 | 11.14 |
| GPTQ (4-Bit) →LoRA | 0.21 | 0.14 | 0.05 | 4.25 | 0.53 | 0.57 | 9.54 |
| GPTQ (8-Bit) →LoRA | 1.0 | 0.72 | 0.06 | 8.25 | 0.56 | 0.61 | 19.99 |
| AWQ (2-Bit) →FT | 0.0 | 0.0 | 0.0 | 2.25 | 0.24 | 0.26 | 33638.44 |
| AWQ (3-Bit) →FT | 1.0 | 0.82 | 0.08 | 3.25 | 0.5 | 0.51 | 7.57 |
| AWQ (4-Bit) →FT | 0.99 | 0.84 | 0.15 | 4.25 | 0.54 | 0.57 | 5.98 |
| AWQ (5-Bit) →FT | 0.99 | 0.85 | 0.12 | 5.25 | 0.55 | 0.6 | 5.68 |
| AWQ (6-Bit) →FT | 0.99 | 0.88 | 0.13 | 6.25 | 0.56 | 0.6 | 5.61 |
| AWQ (8-Bit) →FT | 0.99 | 0.87 | 0.13 | 8.25 | 0.56 | 0.6 | 5.6 |
| AWQ (2-Bit) →MEMIT | 0.0 | 0.0 | 0.0 | 2.25 | 0.24 | 0.26 | 1735678.75 |
| AWQ (3-Bit) →MEMIT | 0.92 | 0.87 | 0.03 | 3.25 | 0.49 | 0.47 | 7.57 |
| AWQ (4-Bit) →MEMIT | 0.92 | 0.89 | 0.04 | 4.25 | 0.55 | 0.58 | 5.97 |
| AWQ (5-Bit) →MEMIT | 0.98 | 0.96 | 0.03 | 5.25 | 0.56 | 0.6 | 5.68 |
| AWQ (6-Bit) →MEMIT | 0.98 | 0.92 | 0.04 | 6.25 | 0.57 | 0.6 | 5.62 |
| AWQ (8-Bit) →MEMIT | 0.98 | 0.94 | 0.04 | 8.25 | 0.57 | 0.6 | 5.59 |
| AWQ (2-Bit) →LoRA | 0.06 | 0.03 | 0.0 | 2.25 | 0.24 | 0.26 | 141960.91 |
| AWQ (3-Bit) →LoRA | 1.0 | 0.71 | 0.05 | 3.25 | 0.5 | 0.51 | 55.44 |
| AWQ (4-Bit) →LoRA | 1.0 | 0.74 | 0.07 | 4.25 | 0.55 | 0.58 | 29.71 |
| AWQ (5-Bit) →LoRA | 1.0 | 0.69 | 0.05 | 5.25 | 0.56 | 0.6 | 17.32 |
| AWQ (6-Bit) →LoRA | 1.0 | 0.69 | 0.06 | 6.25 | 0.57 | 0.61 | 16.6 |
| AWQ (8-Bit) →LoRA | 1.0 | 0.73 | 0.06 | 8.25 | 0.57 | 0.61 | 21.64 |

Table 17: Detailed results for compression → editing.

| | Editing | | | Compression | Unlearning | Utility | |
|---|---|---|---|---|---|---|---|
| | Edit Success | Generalization | Locality | Avg. Bits | Avg. WMDP | MMLU | WikiText PPL |
| GA→SparseGPT (0.25) | 0.0 | 0.0 | 0.0 | 12.25 | 0.46 | 0.51 | inf |
| GA→SparseGPT (0.35) | 0.0 | 0.0 | 0.0 | 10.75 | 0.44 | 0.49 | inf |
| GA→SparseGPT (0.45) | 0.0 | 0.0 | 0.0 | 9.25 | 0.42 | 0.45 | inf |
| GA→SparseGPT (0.55) | 0.0 | 0.0 | 0.0 | 7.75 | 0.3 | 0.32 | inf |
| GA→SparseGPT (0.65) | 0.0 | 0.0 | 0.0 | 6.25 | 0.25 | 0.25 | inf |
| GA→SparseGPT (0.75) | 0.0 | 0.0 | 0.0 | 4.75 | 0.25 | 0.25 | inf |
| GA→Wanda (0.25) | 0.0 | 0.0 | 0.0 | 12.25 | 0.46 | 0.52 | inf |
| GA→Wanda (0.35) | 0.0 | 0.0 | 0.0 | 10.75 | 0.45 | 0.49 | inf |
| GA→Wanda (0.45) | 0.0 | 0.0 | 0.0 | 9.25 | 0.4 | 0.43 | inf |
| GA→Wanda (0.55) | 0.0 | 0.0 | 0.0 | 7.75 | 0.3 | 0.32 | inf |
| GA→Wanda (0.65) | 0.0 | 0.0 | 0.0 | 6.25 | 0.25 | 0.25 | inf |
| GA→Wanda (0.75) | 0.0 | 0.0 | 0.0 | 4.75 | 0.25 | 0.26 | inf |
| GA→GPTQ (2-Bit) | 0.0 | 0.0 | 0.0 | 2.25 | 0.25 | 0.25 | inf |
| GA→GPTQ (3-Bit) | 0.0 | 0.0 | 0.0 | 3.25 | 0.33 | 0.39 | inf |
| GA→GPTQ (4-Bit) | 0.0 | 0.0 | 0.0 | 4.25 | 0.43 | 0.48 | inf |
| GA→GPTQ (8-Bit) | 0.0 | 0.0 | 0.0 | 8.25 | 0.45 | 0.49 | inf |
| GA→AWQ (2-Bit) | 0.0 | 0.0 | 0.0 | 2.25 | 0.25 | 0.25 | 1769133.88 |
| GA→AWQ (3-Bit) | 0.0 | 0.0 | 0.0 | 3.25 | 0.41 | 0.44 | inf |
| GA→AWQ (4-Bit) | 0.0 | 0.0 | 0.0 | 4.25 | 0.43 | 0.47 | inf |
| GA→AWQ (5-Bit) | 0.0 | 0.0 | 0.0 | 5.25 | 0.46 | 0.5 | inf |
| GA→AWQ (6-Bit) | 0.0 | 0.0 | 0.0 | 6.25 | 0.45 | 0.49 | inf |
| GA→AWQ (8-Bit) | 0.0 | 0.0 | 0.0 | 8.25 | 0.45 | 0.49 | inf |
| GD→SparseGPT (0.25) | 0.0 | 0.01 | 0.02 | 12.25 | 0.27 | 0.58 | 5.07 |
| GD→SparseGPT (0.35) | 0.0 | 0.0 | 0.02 | 10.75 | 0.28 | 0.55 | 5.64 |
| GD→SparseGPT (0.45) | 0.0 | 0.0 | 0.02 | 9.25 | 0.28 | 0.51 | 7.02 |
| GD→SparseGPT (0.55) | 0.0 | 0.0 | 0.01 | 7.75 | 0.27 | 0.42 | 10.6 |
| GD→SparseGPT (0.65) | 0.0 | 0.0 | 0.02 | 6.25 | 0.26 | 0.27 | 25.97 |
| GD→SparseGPT (0.75) | 0.01 | 0.0 | 0.01 | 4.75 | 0.24 | 0.23 | 119.71 |
| GD→Wanda (0.25) | 0.0 | 0.01 | 0.03 | 12.25 | 0.35 | 0.58 | 5.34 |
| GD→Wanda (0.35) | 0.0 | 0.0 | 0.02 | 10.75 | 0.38 | 0.56 | 6.61 |
| GD→Wanda (0.45) | 0.0 | 0.0 | 0.01 | 9.25 | 0.46 | 0.51 | 9.9 |
| GD→Wanda (0.55) | 0.0 | 0.0 | 0.0 | 7.75 | 0.37 | 0.39 | 22.22 |
| GD→Wanda (0.65) | 0.0 | 0.0 | 0.0 | 6.25 | 0.26 | 0.23 | 184.75 |
| GD→Wanda (0.75) | 0.0 | 0.0 | 0.0 | 4.75 | 0.26 | 0.23 | 3298.92 |
| GD→GPTQ (2-Bit) | 0.0 | 0.0 | 0.0 | 2.25 | 0.24 | 0.27 | inf |
| GD→GPTQ (3-Bit) | 0.0 | 0.0 | 0.0 | 3.25 | 0.24 | 0.3 | inf |
| GD→GPTQ (4-Bit) | 0.0 | 0.0 | 0.0 | 4.25 | 0.24 | 0.32 | 5.106230513313957e+35 |
| GD→GPTQ (8-Bit) | 0.0 | 0.0 | 0.0 | 8.25 | 0.24 | 0.35 | 2.445586872346403e+26 |
| GD→AWQ (2-Bit) | 0.0 | 0.0 | 0.0 | 2.25 | 0.24 | 0.27 | 2598854.5 |
| GD→AWQ (3-Bit) | 0.0 | 0.0 | 0.0 | 3.25 | 0.33 | 0.33 | inf |
| GD→AWQ (4-Bit) | 0.0 | 0.0 | 0.0 | 4.25 | 0.24 | 0.35 | 2.0452847856930923e+28 |
| GD→AWQ (5-Bit) | 0.0 | 0.0 | 0.0 | 5.25 | 0.25 | 0.36 | 1.0052004503012292e+28 |
| GD→AWQ (6-Bit) | 0.0 | 0.0 | 0.0 | 6.25 | 0.25 | 0.35 | 6.499345951176105e+25 |
| GD→AWQ (8-Bit) | 0.0 | 0.0 | 0.0 | 8.25 | 0.25 | 0.35 | 9.430016153317283e+25 |
| RMU→SparseGPT (0.25) | 0.02 | 0.02 | 0.03 | 12.25 | 0.28 | 0.56 | 5.93 |
| RMU→SparseGPT (0.35) | 0.02 | 0.02 | 0.03 | 10.75 | 0.28 | 0.53 | 6.39 |
| RMU→SparseGPT (0.45) | 0.02 | 0.02 | 0.03 | 9.25 | 0.27 | 0.48 | 8.14 |
| RMU→SparseGPT (0.55) | 0.01 | 0.01 | 0.03 | 7.75 | 0.26 | 0.39 | 10.58 |
| RMU→SparseGPT (0.65) | 0.0 | 0.01 | 0.03 | 6.25 | 0.25 | 0.26 | 35.97 |
| RMU→SparseGPT (0.75) | 0.01 | 0.01 | 0.03 | 4.75 | 0.25 | 0.23 | 101.32 |
| RMU→Wanda (0.25) | 0.02 | 0.02 | 0.04 | 12.25 | 0.29 | 0.56 | 5.88 |
| RMU→Wanda (0.35) | 0.02 | 0.02 | 0.03 | 10.75 | 0.28 | 0.53 | 6.36 |
| RMU→Wanda (0.45) | 0.02 | 0.01 | 0.03 | 9.25 | 0.29 | 0.48 | 7.71 |
| RMU→Wanda (0.55) | 0.01 | 0.01 | 0.03 | 7.75 | 0.27 | 0.35 | 13.73 |
| RMU→Wanda (0.65) | 0.01 | 0.01 | 0.03 | 6.25 | 0.25 | 0.23 | 52.67 |
| RMU→Wanda (0.75) | 0.03 | 0.02 | 0.03 | 4.75 | 0.25 | 0.23 | 334.45 |
| RMU→GPTQ (2-Bit) | 0.0 | 0.01 | 0.03 | 2.25 | 0.25 | 0.24 | 3451.67 |
| RMU→GPTQ (3-Bit) | 0.01 | 0.01 | 0.03 | 3.25 | 0.26 | 0.42 | 9.09 |
| RMU→GPTQ (4-Bit) | 0.02 | 0.02 | 0.03 | 4.25 | 0.27 | 0.53 | 12.88 |
| RMU→GPTQ (8-Bit) | 0.02 | 0.02 | 0.03 | 8.25 | 0.29 | 0.57 | 5.56 |
| RMU→AWQ (2-Bit) | 0.0 | 0.0 | 0.0 | 2.25 | 0.24 | 0.26 | 1726409.12 |
| RMU→AWQ (3-Bit) | 0.01 | 0.01 | 0.03 | 3.25 | 0.27 | 0.45 | 7.55 |
| RMU→AWQ (4-Bit) | 0.02 | 0.02 | 0.03 | 4.25 | 0.29 | 0.56 | 5.95 |
| RMU→AWQ (5-Bit) | 0.02 | 0.03 | 0.04 | 5.25 | 0.28 | 0.56 | 5.68 |
| RMU→AWQ (6-Bit) | 0.02 | 0.02 | 0.03 | 6.25 | 0.28 | 0.56 | 5.62 |
| RMU→AWQ (8-Bit) | 0.02 | 0.02 | 0.03 | 8.25 | 0.29 | 0.57 | 5.56 |

Table 18: Detailed results for unlearning → compression.

| | Editing | | | Compression | Unlearning | | Utility | |
|---|---|---|---|---|---|---|---|
| | Edit Success | Generalization | Locality | Avg. Bits | Avg. WMDP | MMLU | WikiText PPL |
| SparseGPT (0.25) →GA | 0.0 | 0.0 | 0.0 | 12.25 | 0.43 | 0.45 | inf |
| SparseGPT (0.35) →GA | 0.0 | 0.0 | 0.0 | 10.75 | 0.34 | 0.36 | inf |
| SparseGPT (0.45) →GA | 0.0 | 0.0 | 0.0 | 9.25 | 0.31 | 0.33 | inf |
| SparseGPT (0.55) →GA | 0.0 | 0.0 | 0.0 | 7.75 | 0.25 | 0.25 | inf |
| SparseGPT (0.65) →GA | 0.0 | 0.0 | 0.0 | 6.25 | 0.27 | 0.28 | inf |
| SparseGPT (0.75) →GA | 0.01 | 0.01 | 0.03 | 4.75 | 0.26 | 0.23 | inf |
| SparseGPT (0.25) →GD | 0.01 | 0.0 | 0.02 | 12.25 | 0.5 | 0.48 | 13.32 |
| SparseGPT (0.35) →GD | 0.01 | 0.01 | 0.01 | 10.75 | 0.29 | 0.25 | 2712.26 |
| SparseGPT (0.45) →GD | 0.0 | 0.0 | 0.0 | 9.25 | 0.25 | 0.48 | 1.246813090597988e+19 |
| SparseGPT (0.55) →GD | 0.0 | 0.0 | 0.02 | 7.75 | 0.35 | 0.36 | 9.35 |
| SparseGPT (0.65) →GD | 0.0 | 0.0 | 0.01 | 6.25 | 0.24 | 0.26 | inf |
| SparseGPT (0.75) →GD | 0.0 | 0.01 | 0.01 | 4.75 | 0.26 | 0.23 | 1437619072.0 |
| SparseGPT (0.25) →RMU | 0.01 | 0.02 | 0.04 | 12.25 | 0.31 | 0.57 | 5.92 |
| SparseGPT (0.35) →RMU | 0.0 | 0.02 | 0.02 | 10.75 | 0.31 | 0.54 | 6.35 |
| SparseGPT (0.45) →RMU | 0.01 | 0.01 | 0.03 | 9.25 | 0.31 | 0.51 | 7.86 |
| SparseGPT (0.55) →RMU | 0.01 | 0.01 | 0.03 | 7.75 | 0.29 | 0.41 | 10.39 |
| SparseGPT (0.65) →RMU | 0.0 | 0.0 | 0.02 | 6.25 | 0.26 | 0.24 | 30.26 |
| SparseGPT (0.75) →RMU | 0.01 | 0.01 | 0.02 | 4.75 | 0.25 | 0.23 | 97.03 |
| Wanda (0.25) →GA | 0.0 | 0.0 | 0.0 | 12.25 | 0.49 | 0.54 | inf |
| Wanda (0.35) →GA | 0.0 | 0.0 | 0.0 | 10.75 | 0.44 | 0.51 | inf |
| Wanda (0.45) →GA | 0.0 | 0.0 | 0.0 | 9.25 | 0.41 | 0.43 | inf |
| Wanda (0.55) →GA | 0.0 | 0.0 | 0.0 | 7.75 | 0.32 | 0.34 | inf |
| Wanda (0.65) →GA | 0.0 | 0.0 | 0.0 | 6.25 | 0.26 | 0.26 | inf |
| Wanda (0.75) →GA | 0.01 | 0.01 | 0.03 | 4.75 | 0.26 | 0.23 | 8.12597183809491e+36 |
| Wanda (0.25) →GD | 0.0 | 0.0 | 0.0 | 12.25 | 0.52 | 0.57 | 13.26 |
| Wanda (0.35) →GD | 0.0 | 0.0 | 0.0 | 10.75 | 0.45 | 0.48 | 57480503296.0 |
| Wanda (0.45) →GD | 0.0 | 0.0 | 0.0 | 9.25 | 0.5 | 0.53 | 49.77 |
| Wanda (0.55) →GD | 0.0 | 0.0 | 0.0 | 7.75 | 0.39 | 0.35 | 121.07 |
| Wanda (0.65) →GD | 0.0 | 0.0 | 0.0 | 6.25 | 0.26 | 0.23 | 16268.01 |
| Wanda (0.75) →GD | 0.0 | 0.0 | 0.0 | 4.75 | 0.25 | 0.25 | 10226426.0 |
| Wanda (0.25) →RMU | 0.02 | 0.02 | 0.04 | 12.25 | 0.32 | 0.57 | 5.86 |
| Wanda (0.35) →RMU | 0.02 | 0.02 | 0.03 | 10.75 | 0.34 | 0.56 | 6.33 |
| Wanda (0.45) →RMU | 0.02 | 0.02 | 0.03 | 9.25 | 0.39 | 0.52 | 7.55 |
| Wanda (0.55) →RMU | 0.01 | 0.01 | 0.03 | 7.75 | 0.36 | 0.36 | 12.58 |
| Wanda (0.65) →RMU | 0.01 | 0.01 | 0.02 | 6.25 | 0.26 | 0.23 | 45.53 |
| Wanda (0.75) →RMU | 0.02 | 0.03 | 0.02 | 4.75 | 0.26 | 0.23 | 391.87 |
| GPTQ (2-Bit) →GA | 0.0 | 0.0 | 0.0 | 2.25 | 0.25 | 0.25 | inf |
| GPTQ (3-Bit) →GA | 0.0 | 0.0 | 0.0 | 3.25 | 0.24 | 0.27 | inf |
| GPTQ (4-Bit) →GA | 0.0 | 0.0 | 0.0 | 4.25 | 0.34 | 0.38 | inf |
| GPTQ (8-Bit) →GA | 0.0 | 0.0 | 0.0 | 8.25 | 0.45 | 0.5 | inf |
| GPTQ (2-Bit) →GD | 0.0 | 0.0 | 0.0 | 2.25 | 0.25 | 0.25 | inf |
| GPTQ (3-Bit) →GD | 0.0 | 0.0 | 0.0 | 3.25 | 0.26 | 0.24 | 628.24 |
| GPTQ (4-Bit) →GD | 0.01 | 0.01 | 0.01 | 4.25 | 0.27 | 0.56 | 9.38 |
| GPTQ (8-Bit) →GD | 0.01 | 0.02 | 0.05 | 8.25 | 0.26 | 0.47 | inf |
| GPTQ (2-Bit) →RMU | 0.0 | 0.0 | 0.0 | 2.25 | 0.25 | 0.24 | 1017825.94 |
| GPTQ (3-Bit) →RMU | 0.01 | 0.01 | 0.02 | 3.25 | 0.32 | 0.32 | 11.1 |
| GPTQ (4-Bit) →RMU | 0.0 | 0.02 | 0.02 | 4.25 | 0.45 | 0.58 | 6.35 |
| GPTQ (8-Bit) →RMU | 0.01 | 0.02 | 0.03 | 8.25 | 0.28 | 0.57 | 5.58 |
| AWQ (2-Bit) →GA | 0.0 | 0.0 | 0.0 | 2.25 | 0.24 | 0.27 | inf |
| AWQ (3-Bit) →GA | 0.0 | 0.0 | 0.0 | 3.25 | 0.24 | 0.27 | inf |
| AWQ (4-Bit) →GA | 0.0 | 0.0 | 0.0 | 4.25 | 0.45 | 0.45 | inf |
| AWQ (5-Bit) →GA | 0.0 | 0.0 | 0.0 | 5.25 | 0.35 | 0.39 | inf |
| AWQ (6-Bit) →GA | 0.0 | 0.0 | 0.0 | 6.25 | 0.25 | 0.3 | inf |
| AWQ (8-Bit) →GA | 0.0 | 0.0 | 0.0 | 8.25 | 0.28 | 0.37 | inf |
| AWQ (2-Bit) →GD | 0.0 | 0.0 | 0.02 | 2.25 | 0.24 | 0.27 | 4.496028624899119e+33 |
| AWQ (3-Bit) →GD | 0.01 | 0.0 | 0.02 | 3.25 | 0.29 | 0.34 | 166895.91 |
| AWQ (4-Bit) →GD | 0.01 | 0.0 | 0.04 | 4.25 | 0.24 | 0.42 | 7.05 |
| AWQ (5-Bit) →GD | 0.0 | 0.0 | 0.03 | 5.25 | 0.25 | 0.38 | 2685384.0 |
| AWQ (6-Bit) →GD | 0.0 | 0.0 | 0.05 | 6.25 | 0.25 | 0.31 | 27025.07 |
| AWQ (8-Bit) →GD | 0.0 | 0.0 | 0.01 | 8.25 | 0.25 | 0.25 | 49224794112.0 |
| AWQ (2-Bit) →RMU | 0.0 | 0.0 | 0.0 | 2.25 | 0.24 | 0.27 | 1749321.75 |
| AWQ (3-Bit) →RMU | 0.01 | 0.01 | 0.03 | 3.25 | 0.27 | 0.46 | 7.51 |
| AWQ (4-Bit) →RMU | 0.02 | 0.02 | 0.03 | 4.25 | 0.27 | 0.55 | 6.06 |
| AWQ (5-Bit) →RMU | 0.02 | 0.03 | 0.04 | 5.25 | 0.27 | 0.56 | 5.68 |
| AWQ (6-Bit) →RMU | 0.02 | 0.02 | 0.04 | 6.25 | 0.28 | 0.56 | 5.6 |
| AWQ (8-Bit) →RMU | 0.02 | 0.03 | 0.03 | 8.25 | 0.28 | 0.57 | 5.71 |

Table 19: Detailed results for compression → unlearning.

| | Editing | | | Compression | Unlearning | Utility | |
|---|---|---|---|---|---|---|---|
| | Edit Success | Generalization | Locality | Avg. Bits | Avg. WMDP | MMLU | WikiText PPL |
| FT→GA | 0.0 | 0.0 | 0.0 | 16.0 | 0.47 | 0.53 | inf |
| FT→GD | 0.32 | 0.25 | 0.07 | 16.0 | 0.29 | 0.41 | 2.8236466916108585e+31 |
| FT→RMU | 0.99 | 0.82 | 0.1 | 16.0 | 0.28 | 0.56 | 5.61 |
| MEMIT→GA | 0.0 | 0.0 | 0.0 | 16.0 | 0.4 | 0.45 | inf |
| MEMIT→GD | 0.53 | 0.49 | 0.03 | 16.0 | 0.26 | 0.36 | 27645056122880.0 |
| MEMIT→RMU | 0.96 | 0.89 | 0.03 | 16.0 | 0.29 | 0.56 | 5.58 |
| LoRA→GA | 0.0 | 0.0 | 0.0 | 16.0 | 0.28 | 0.29 | inf |
| LoRA→GD | 0.44 | 0.23 | 0.05 | 16.0 | 0.3 | 0.45 | 29.46 |
| LoRA→RMU | 1.0 | 0.68 | 0.05 | 16.0 | 0.3 | 0.52 | 34.56 |

Table 20: Detailed results for editing → unlearning.

| | Editing | | | Compression | Unlearning | Utility | |
|---|---|---|---|---|---|---|---|
| | Edit Success | Generalization | Locality | Avg. Bits | Avg. WMDP | MMLU | WikiText PPL |
| GA→FT | 0.07 | 0.04 | 0.0 | 16.0 | 0.47 | 0.52 | inf |
| GA→MEMIT | 0.48 | 0.41 | 0.0 | 16.0 | 0.45 | 0.49 | inf |
| GA→LoRA | 1.0 | 0.78 | 0.03 | 16.0 | 0.34 | 0.36 | 56.91 |
| GD→FT | 0.99 | 0.81 | 0.11 | 16.0 | 0.29 | 0.59 | 4.64 |
| GD→MEMIT | 0.93 | 0.89 | 0.05 | 16.0 | 0.28 | 0.58 | 4.65 |
| GD→LoRA | 1.0 | 0.71 | 0.08 | 16.0 | 0.54 | 0.59 | 4.99 |
| RMU→FT | 1.0 | 0.79 | 0.13 | 16.0 | 0.32 | 0.57 | 5.6 |
| RMU→MEMIT | 0.97 | 0.93 | 0.03 | 16.0 | 0.3 | 0.56 | 5.58 |
| RMU→LoRA | 1.0 | 0.71 | 0.05 | 16.0 | 0.29 | 0.56 | 12.83 |

Table 21: Detailed results for unlearning → editing.

