# OpenReview forum: "Composable Interventions for Language Models"
_ICLR.cc/2025/Conference — ICLR 2025 Poster_

### Official Review · Reviewer_5FNv · 2024-11-01

**Soundness:** 2
**Presentation:** 3
**Contribution:** 2
**Rating:** 3
**Confidence:** 5

**Summary:**

The paper highlights a gap in the practical application of model interventions, where multiple techniques, such as information removal, model editing, and efficiency optimization (e.g., pruning and quantization) are often needed in combination but have largely been studied in isolation. The authors systematically explore how these interventions interact when used together, analyzing the sensitivity to intervention order and the broader implications of composability. This research offers insights into the outcome of utilizing multiple interventions and emphasizes the need for multi-objective intervention metrics to gauge their effective composability.

**Strengths:**

The research effectively highlights an oversight in current evaluation methodologies, where interventions are assessed independently despite concurrent deployment requirements in productionized environments.

Analysis of compositional effects across multiple intervention types, incorporating various established techniques from each domain (editing, unlearning, and model compression).

Findings provide valuable insights into the compatibility and limitations of different intervention combinations, offering practical usage guidance.

**Weaknesses:**

While the study provides a thorough documentation of interactions among various interventions, it falls short in examining the underlying issues or mechanisms that lead to compositional failures. A deeper analysis of the interference patterns or cause-effect relationships between interventions is necessary to understand why the composition of approaches works more effectively in some cases than in others. Exploring why certain techniques exhibit superior composability over others would greatly enhance the study’s contribution. Such insights would also help clarify the conditions under which specific interventions interact constructively or destructively, offering guidance for future research in designing more robust and composable interventions.

**Questions:**

How do different interventions interact with one another? Are there any interference patterns or dependencies that affect their compositional effectiveness?

Why do certain techniques exhibit better composability than others? Are there particular properties or design choices in those techniques that lend themselves to more effective composition?

---

> ### Author Response · Authors · 2024-11-25
> **Response**
>
> Thank you for your thoughtful review and for recognising the contributions of our work in addressing gaps in current evaluation methodologies. We are particularly grateful for your acknowledgment of the practical relevance of our analysis of compositional effects and the insights our findings provide for real-world application.
>
> We hope that our responses and updates address your concerns and encourage you to consider raising your score to calibrate with the other reviewers, who notice a similar weakness but are overall positive.  If you still have concerns about our work, we would greatly appreciate further guidance on how we can improve our work to achieve a better score in your review.
> ### W: Examining the underlying mechanisms and causes of composability differences.
> We do not believe this is clearly a weakness of the work we do present in the paper—it is a broad suggestion that more work can be done on this topic. We absolutely agree that our work motivates deeper analysis of the underlying mechanisms behind compositional failures. We believe our work can clearly inspire such analyses. But taken together, the breadth of questions our work suggests, the pace of new intervention development, and the pace of model releases suggests that such analysis is best conducted across the community and by researchers during intervention development. Our work is a necessary step towards that goal.
>
> Our findings also already offer some guidance for future research by identifying which techniques exhibit superior composability and under what conditions interventions tend to interact constructively or destructively. For example, our results reveal previously-unknown patterns, such as compression hindering editability and the compatibility of unlearning and editing, which are critical for guiding immediate application and future design.
>
> Overall, our work lays a necessary foundation to showcase that composability is already an issue and to encourage the research community to consider composability as a metric during intervention design. In fact, composability can already guide practitioners as a validation metric during method development and hyperparameter tuning.
>
> ### Q1: How do different interventions interact with one another? Are there any interference patterns or dependencies that affect their compositional effectiveness?
>  We study interactions between interventions in terms of resultant model performance. Through this lens, there are many ways interventions interact:
> Order Dependence: As demonstrated by the Order Sensitivity metric, the sequence in which interventions are applied can significantly impact their effectiveness. For example, compression applied before editing tends to better preserve editability, whereas the reverse often leads to degraded outcomes. Similarly, certain unlearning methods perform more robustly when applied before compression.
> Category-Specific Interactions: Different categories of interventions exhibit unique interaction dynamics. Compression methods often hinder the success of subsequent editing or unlearning, likely due to changes in how knowledge is encoded. In contrast, editing and unlearning show higher compatibility, suggesting that their objectives and mechanisms are less conflicting.
> Technique-Specific Interactions: Certain methods, such as AWQ for compression and RMU for unlearning, consistently exhibit superior composability across different setups. These findings highlight the inherent differences in how techniques interfere with or complement one another.
> ### Q2: Why do certain techniques exhibit better composability than others? Are there particular properties or design choices in those techniques that lend themselves to more effective composition?
> Understanding why certain techniques exhibit better composability is an exciting direction. While this is future work, we hypothesize that factors such as localized impact, objective alignment, and robustness to interference play a significant role. For example,  interventions specifically designed to minimize disruption to unrelated model behavior, like RMU and MEMIT, tend to compose more effectively with other interventions.

---

> > ### Author Response · Authors · 2024-11-27
> > **Kind reminder of our response**
> >
> > Thank you again for your feedback and your help improving our work! We'd like to kindly remind you that we've addressed your concerns. As the discussion period closes, we'd love to know if you have any lingering concerns that we can address within the timeframe.

---

> > > ### Comment · Reviewer_5FNv · 2024-12-02
> > > **Response Reviewer**
> > >
> > > >  For example, compression applied before editing tends to better preserve editability, whereas the reverse often leads to degraded outcomes.
> > >
> > > Compression introduces performance limitations, with a direct trade-off between compression percentage and model capabilities. The performance degradation appears predictable.
> > >
> > > >  Similarly, certain unlearning methods perform more robustly when applied before compression.
> > >
> > > Unlearning effectiveness varies by approach. Parameter shifts during unlearning could potentially be mitigated by compression, particularly when parameters are subsequently quantized or pruned. A deeper analysis is required to comprehensively understand these interactions.
> > >
> > > > Compression methods often hinder the success of subsequent editing or unlearning, likely due to changes in how knowledge is encoded.
> > >
> > > Table 11 indicates performance decline correlates with compression extent. Reduced fine-tuning performance suggests broader model trainability challenges rather than composability.
> > >
> > > My final comment is that the work's primary contribution is a framework that highlights systemic challenges in intervention interactions, which necessitates rigorous, granular reasoning to substantiate its claims.

---

### Official Review · Reviewer_1gkW · 2024-11-02

**Soundness:** 3
**Presentation:** 3
**Contribution:** 3
**Rating:** 6
**Confidence:** 4

**Summary:**

This work introduces a "Composable Interventions" for language models, addressing a practical, real-world need. The authors develop new metrics and a unified codebase to evaluate the effectiveness of combining multiple interventions, such as knowledge editing, model compression, and machine unlearning. Through rigorous and extensive experimentation with 417 intervention combinations, the research underscores the significance of the sequence in which interventions are applied on model performance. The exposition is clear and well-organized. This study tackles the complexities of applying multiple interventions simultaneously, providing foundational insights that pave the way for future research on robust and adaptable language models.

**Strengths:**

- The authors introduces two new metrics:Order-free Error and Order Sensitivity, that evaluate the robustness of intervention sequences, highlighting the importance of intervention order.
- The authors also conducted extensive experiments using various compositions, showcasing the interaction effects of various interventions.
- The authors present several important findings. For instance, the success of interventions is highly dependent on the order of application; compression hinders editability; unlearning and editing are relatively compatible; and certain methods are inherently more suitable for sequential applications. Additionally, they found that conventional metrics are inadequate for measuring composability, underscoring the need for new intervention techniques explicitly designed for this purpose.

**Weaknesses:**

- The authors present numerous findings and insights, such as Compression consistently hinders Knowledge Editing and Unlearning, which is a relatively straightforward conclusion. It would be valuable for the authors to explore deeper analysis and hypotheses beyond identifying patterns such as 'A affects B' or 'A and B are compatible,' moving from experimental observations to more comprehensive insights.
- In combined interventions, it appears that certain combinations do not yet have a clear consensus. For instance, in composing model compression with knowledge editing, the preferred order can vary significantly by method: for GPTQ, editing seems more effective first, whereas for AWQ, compression may be preferable. This suggests that optimal ordering may rely on experimental results specific to each method.
- While Order-free Error and Order Sensitivity offer valuable insights, the small differences in outcomes and variations across metrics suggest that additional measures may be needed to fully capture intervention interactions.

**Questions:**

Refer to Weaknesses

---

> ### Author Response · Authors · 2024-11-25
> **Response**
>
> Thank you for your thoughtful feedback and for recognising the contributions of our work. We sincerely appreciate your positive assessment of our new metrics, the extensive experimental analysis, and the key findings that highlight the significance of intervention order and composability.
>
> In response to your comments, we have carefully addressed the points you raised. These updates aim to further clarify our contributions and strengthen the paper, building on the insights you highlighted in your review.
>
> We hope that our revisions and detailed responses adequately address your concerns, and we kindly invite you to consider raising your score. If there are additional areas where we could improve, we would greatly value your guidance. Thank you for your constructive feedback and encouragement.
> ### W1: Including more comprehensive insights
> We acknowledge that identifying patterns like compression's impact on editing is just a first step. While deeper analysis is an exciting direction for future work, this work intentionally focuses on establishing the foundational evidence for composability as a critical consideration. The rapid pace of model development makes such detailed analysis better suited for broader community investigation. Our findings already provide practical guidance while demonstrating composability's value as a validation metric during intervention development.
> ### W2: Optimal composability may rely on each method
> This is not a practical weakness of our work as such but an interesting finding that has implications for researchers and practitioners. It is true that methods within the same intervention class do not always have the same composability as other methods within the same intervention class. The consequence of this is that researchers working on new methods within a given class should test the composability of their new method for completeness. Beyond identifying such findings, our framework provides a standardized way for them to do this.
>
> However, while we acknowledge outliers, there are general trends of composability between intervention classes that give valuable insights: *edit before compressing* and *unlearn before compressing* are two such trends.
> ### W3: Low fidelity in results using order-free error and order sensitivity
> Our metrics indeed can be similar between methods from within the same class. While we agree they provide valuable insights already,  we also agree there may still be more dimensions to examine in the future. However, we also note that we extend beyond the composability metrics to also derive insights from intervention-specific metrics (e.g., Figures 2 and 3). By pairing intervention-specific and composability metrics, we believe our experiments capture sufficient intervention interactions for one paper.
>
> We’d also like to clarify that small differences in the composability metrics successfully reflect small differences in the full interactions (comparing Figures 2 and 3 with their corresponding tables 2 and 3). And the order-free error and order sensitivity successfully capture wide differences in composability where they exist. Two examples of this are: 1) the edit success order sensitivity of LoRA when composed with different unlearning methods (Table 4) ranges from 0.00 to 1.00, using the full fidelity of the metric, and 2) edit success order-free error when composing knowledge editing with compression (Table 2) ranges from 0.10 to 0.59.

---

> > ### Comment · Reviewer_1gkW · 2024-11-26
> > **Response to Submission7753 Authors**
> >
> > Thanks for the clarifications. I will keep my positive score.

---

### Official Review · Reviewer_gvGR · 2024-11-02

**Soundness:** 2
**Presentation:** 3
**Contribution:** 3
**Rating:** 6
**Confidence:** 4

**Summary:**

The paper proposes a framework (evaluation metrics) to study the effect of the composable interventions on LLMs. Authors claim that interventions like model compression, knowledge editing, and unlearning can affect each other and hence they study this phenomenon.

**Strengths:**

1. The paper is written in a very clear, nice, and easy to understand way.
2. The motivation of the paper is clear.
3. The paper studies an interesting problem.

**Weaknesses:**

While the paper is written very clearly and it studies an interesting problem, I have a major concern. My major concern is that the technical contribution of the paper might not be rigorous although insights and findings are good and the paper studies an interesting problem. perhaps one way to improve the paper would be to propose a method to make LLMs more robust to composable interventions or even design the interventions themselves in a way to not see degrade in their performance after other interventions are applied on top. Or even using the proposed evaluation framework as a recommendation framework to provide recommendation on how the interventions should be applied based on a given LLM and the intervention method.

Another concern I have is that the framework might be limited as it only considers two interventions at a time. Can the method be expanded to study more than two interventions at a time?

Finally, results on other LLMs are not fully consistant across all the studied dimensions which makes the finding of the paper not super reliable. It would be good to do some statistical significance test (e.g., across various LLMs).

**Questions:**

1. Why did you decide to focus on the three interventions in particular mentioned in the paper? (unlearning, compressing, model editing)
Would this choice limit the scope of study?

2. As also mentioned in the weaknesses above, the framework might be limited as it only considers two interventions at a time. Can the method be expanded to study more than two interventions at a time?

---

> ### Author Response · Authors · 2024-11-25
> **Response**
>
> Thank you for your thoughtful feedback and for highlighting the clarity of our writing, the strength of our motivation, and the importance of the problem we address. We greatly appreciate your recognition of these aspects of our work. We have thoroughly considered your comments and have conducted additional experiments and clarified key points in the manuscript to address your concerns. We believe these revisions strengthen the paper and provide further insight into the problem and our proposed solution.
>
> We hope that our detailed responses and updates address your concerns, and we kindly encourage you to consider raising your score. If any questions or uncertainties remain, we would greatly appreciate further guidance on how we might improve our work to meet your expectations. Thank you again for your valuable feedback.
> ### (Major Concern) W1: No new technique.
> We believe our insights and findings are sufficient to warrant publication—the ICLR call for papers explicitly calls for “datasets and benchmarks” as a relevant topic. Our contributions match comparable papers, which rarely include technical contributions. In fact, we think the scale of our experiments on an interesting and novel problem successfully lays the necessary foundations for future research in this area. So indeed, we propose no new technique (beyond novel measures of intervention composability). But we don’t agree this is a *major* weakness.
>
> For providing recommendations, this is a great suggestion. The majority of our results section can be viewed as a set of recommendations. To strengthen this point and clarify for future readers, we have distilled our recommendations into the following table which will be displayed predominantly in Section 4 of our paper. Thanks for this suggestion!
>
> | Category | Best-Performing Composition | Analysis |
> |----------|---------------------------|-----------|
> | Editing & Compression | AWQ→MEMIT | Section 4.1, Figure 2 |
> | Unlearning & Compression | RMU→AWQ | Section 4.2, Figure 3 |
> | Editing & Unlearning | RMU↔MEMIT | Section 4.3, Table 4 |
>
> ### W2: Extending to more than two interventions
> Our framework absolutely extends to more than two interventions. While we can’t exhaustively consider all combinations of three interventions, we have taken your suggestion and run new experiments composing three interventions. We limit the search space by choosing the most-composable intervention from each category: WANDA 25% sparsity (compression), MEMIT (editing), and RMU (unlearning). Expanding to three interventions allows us to ask the following research questions:
>
> Research Question: Does composing more than two interventions lead to catastrophic regression in overall performance?
> Across base and instruction-tunes Llama-3 8B, we find that composing these three interventions does not lead to significant variance in overall performance as measured by MMLU. These results are in line with our observations from our initial submission where we find little variance in MMLU performance across compositions in contrast to at times extreme variance in intervention-specific metrics. We also observe similar trends in composability across base and instruct models with triplet compositions.
>
> **Llama-3 8B Base**
> | Intervention 1 | Intervention 2 | Intervention 3 | MMLU | Avg WMDP | Edit Success | Edit Generalization | Avg Success + Generalization |
> |--|----|-----|------|-----------|-----|------|---|
> | WANDA | RMU | MEMIT | 56.36% | 30.77% | 99.20% | 93.70% | 96.45% |
> | MEMIT | WANDA | RMU | 54.63% | 27.60% | 99.50% | 90.30% | 94.90% |
> | MEMIT | RMU | WANDA | 54.17% | 28.00% | 99.50% | 90.12% | 94.81% |
> | WANDA | MEMIT | RMU | 55.23% | 31.30% | 95.86% | 90.26% | 93.06% |
> | RMU | WANDA | MEMIT | 53.87% | 28.20% | 93.00% | 92.00% | 92.50% |
> | RMU | MEMIT | WANDA | 54.32% | 28.20% | 94.67% | 85.17% | 89.92% |
>
> **Llama-3 8B Instruct**
> | Intervention 1 | Intervention 2 | Intervention 3 | MMLU | Avg WMDP | Edit Success | Edit Generalization | Avg Success + Generalization |
> |---|---|-----|------|-----------|---------|---|---|
> | WANDA | MEMIT | RMU | 59.09% | 34.68% | 99.10% | 84.77% | 91.93% |
> | MEMIT | WANDA | RMU | 54.27% | 26.98% | 94.60% | 85.27% | 89.93% |
> | MEMIT | RMU | WANDA | 54.61% | 27.39% | 91.53% | 80.47% | 86.00% |
> | WANDA | RMU | MEMIT | 57.57% | 31.46% | 86.72% | 83.50% | 85.11% |
> | RMU | WANDA | MEMIT | 53.77% | 26.41% | 86.00% | 80.77% | 83.39% |
> | RMU | MEMIT | WANDA | 54.53% | 26.33% | 74.82% | 81.44% | 78.13% |
>
> ### W3: Inconsistencies in composability patterns across models
> We believe this is not inherently a weakness of our work, but a weakness of the popular models and interventions we consider—composability is currently worse than expected and less-predictable than we’d like. This is itself an important finding, not a foundational flaw of our work. We hope our work can especially inspire future intervention developers to at least consider composability as a validation metric to begin rectifying this issue.

---

> > ### Comment · Reviewer_gvGR · 2024-11-25
> > **Response to the Authors**
> >
> > I thank the authors for their responses and the addition of experiments. I also read other reviewers comments and I agree with them in that the generalizability of some results might not be clear specially that some of the initial existing results were not completely consistant as I also mentioned in my own review. With that being said, the paper has other merits that I really appreciate it and I also appreciate the addition of results that authors have provided to strengthen their paper. If some methodological contribution was also included in the paper, i would have for sure increased my score. For now, I will keep my score; however, I am open to discussion with my peer reviewers.

---

> > > ### Comment · Reviewer_Mrdn · 2024-11-27
> > >
> > > Thanks for the clarification and uploading the code! I will keep my positive score.

---

> > > > ### Comment · Reviewer_gvGR · 2024-11-27
> > > > **Response to Reviewer  Mrdn from Reviewer gvGR**
> > > >
> > > > Reviewer Mrdn, I think you posted your response into the wrong box :) This is my review (reviewer gvGR) :)

---

### Official Review · Reviewer_Mrdn · 2024-11-03

**Soundness:** 2
**Presentation:** 3
**Contribution:** 3
**Rating:** 6
**Confidence:** 3

**Summary:**

This paper proposes a new framework to assess the effects of combining different test-time interventions on language models to improve factual accuracy, efficiency, and mitigate harmful outputs. Traditionally, methods such as knowledge editing, model compression, and machine unlearning have been developed separately, but practical applications often require these interventions to work together. The authors introduce two novel metrics—Order-free Error and Order Sensitivity—to evaluate the composability of interventions, aiming to ensure that each intervention's success is minimally impacted by other interventions.

Key contributions include: (1) extensive experiments across 417 combinations of interventions, uncovering findings, such as compression often hindering the success of knowledge editing and unlearning, and the significant impact of intervention order on outcomes. (2) They will provide a codebase featuring popular intervention methods, supporting future multi-objective intervention research and development.
The paper emphasizes the need for composable, multi-objective intervention methods and creates a foundation for systematically evaluating intervention interactions in language models.

**Strengths:**

* Addresses an important program in model updates and adaptations to real-world requirements.
Introduces new metrics and provides extensive experimental results.
* Codebase hypothetically supports is flexible enough to scale to other inference-time interventions. There is no code uploaded, so hard to say for sure.
* Well-written paper

**Weaknesses:**

* Hard to say if the findings using the specific models and datasets used generalize more broadly (e.g., Llama 3, WMDP). WMDP, for instance, specifically notes that “benchmarking on only WMDP may yield a false sense of model safety after unlearning.”
* The paper lacks guidelines for ordering interventions or insights/analysis into why some orderings are not robust or how to make them more robust.

**Questions:**

See weaknesses.

---

> ### Author Response · Authors · 2024-11-25
> **Response**
>
> Thank you for your thorough review! We are glad that you believe we study an important problem, that our experiments capture the problem extensively , and that our framework is flexible to scale and adapt new interventions . We respond to your suggestions as follows:
>
> ### W1: Generalization to other models:
> We’d like to clarify that our experiments already span Llama, Mistral [1], and Yi [2] models (Table 8). We have also extended our experiments to include Llama-3-8b-instruct and Llama-3.2-1b (see general response). We will make this clearer by moving these experiments into the main paper. In these experiments, we find that  composability results do generalize across these LMs.
>
> Regarding evaluation datasets, we agree that it is an interesting direction to better understand how far our findings generalize, especially to other datasets. As you note, every dataset has flaws.  By studying popular models, datasets, and interventions, our work is a sufficient first step. We will further clarify in the conclusion of the paper that expanding our study to future other datasets is a promising direction for future work.
>
> ### W2.1: Lack of guidelines
> This is a very useful suggestion which will help improve our work by showcasing some key findings. For the interventions we study, we share the best-performing interventions and link to our in-depth analysis in the below table. We will add this concise summary of our key takeaways to Section 4 of our paper.
>
> | Category | Best-Performing Composition | Analysis |
> |----------|---------------------------|-----------|
> | Editing & Compression | AWQ→MEMIT | Section 4.1, Figure 2 |
> | Unlearning & Compression | RMU→AWQ | Section 4.2, Figure 3 |
> | Editing & Unlearning | RMU↔MEMIT | Section 4.3, Table 4 |
>
>
> ### W2.2: Determining Root Cause of Composability
>
> We agree that better understanding the mechanisms within each intervention that improves or regresses composability is an interesting and important direction. We will expand our conclusions to include some possible factors for composability, building on recent works. For example, recent works in unlearning have highlighted that today’s best approximate unlearning techniques may largely obfuscate knowledge rather than robustly unlearning knowledge from the LM’s weights [5, 6]. While answering these questions is beyond the scope of this current work, further discussion of possible factors in intervention implementations which influence composability can increase our work’s impact as a catalyst for future work on composability.
>
> ### No code uploaded
> We’d like to clarify that our code actually is uploaded in an anonymous repository https://anonymous.4open.science/r/composable-interventions-D005/README.md (see L53 on Page 1). Failing to also upload our code as supplemental materials was an oversight on our part, thank you for pointing this out.
>
> ### Citations
> [1] Young, 0.A., Chen, B., Li, C., Huang, C., Zhang, G., Zhang, G., Li, H., Zhu, J., Chen, J., Chang, J., Yu, K., Liu, P., Liu, Q., Yue, S., Yang, S., Yang, S., Yu, T., Xie, W., Huang, W., Hu, X., Ren, X., Niu, X., Nie, P., Xu, Y., Liu, Y., Wang, Y., Cai, Y., Gu, Z., Liu, Z., & Dai, Z. (2024). Yi: Open Foundation Models by 01.AI. ArXiv, abs/2403.04652.
>
> [2] Jiang, A.Q., Sablayrolles, A., Mensch, A., Bamford, C., Chaplot, D.S., Casas, D.D., Bressand, F., Lengyel, G., Lample, G., Saulnier, L., Lavaud, L.R., Lachaux, M., Stock, P., Scao, T.L., Lavril, T., Wang, T., Lacroix, T., & Sayed, W.E. (2023). Mistral 7B. ArXiv, abs/2310.06825.
>
> [3] Li, N., Han, Z., Steneker, I., Primack, W., Goodside, R., Zhang, H., Wang, Z., Menghini, C., & Yue, S. (2024). LLM Defenses Are Not Robust to Multi-Turn Human Jailbreaks Yet. ArXiv, abs/2408.15221.
>
> [4] Thaker, P., Hu, S., Kale, N., Maurya, Y., Wu, Z.S., & Smith, V. (2024). Position: LLM Unlearning Benchmarks are Weak Measures of Progress. ArXiv, abs/2410.02879.
>
> [5] Deeb, A., & Roger, F. (2024). Do Unlearning Methods Remove Information from Language Model Weights?
>
> [6] Lucki, J., Wei, B., Huang, Y., Henderson, P., Tramèr, F.S., & Rando, J. (2024). An Adversarial Perspective on Machine Unlearning for AI Safety.

---

> > ### Author Response · Authors · 2024-11-27
> > **Kind reminder of our response**
> >
> > Thank you again for your feedback and your help improving our work! We'd like to kindly remind you that we've addressed your concerns. As the discussion period closes, we'd love to know if you have any lingering concerns that we can address within the timeframe.

---

### Official Review · Reviewer_KMdD · 2024-11-04

**Soundness:** 3
**Presentation:** 3
**Contribution:** 3
**Rating:** 8
**Confidence:** 4

**Summary:**

This paper experimentally demonstrates the effect of composing various inference time setups for LLMs and suggests possible setups for better performance from its analysis.

**Strengths:**

S1. The paper conducts experiments on various methods and compositions on multiple models to provide a deeper insight.

S2. The method proposes metrics that correlate with the effect of a composition.

S3. The presentation is lucid and easy to follow.

S4. The codebase will be useful to the community for further studies.

**Weaknesses:**

W1. Some comparisons on the base models, chat, and RLHF could be interesting which could provide insight into pretraining, instruction tuning, and post-training with the interventions.

W2. With a similar spirit as W1, it is important to see the results for different generations of the same model family and sizes.

W3. The sensitivity (standard deviation) of the experiments is unclear if run multiple times in Table 2.

W4. Practitioners need to perform a grid search for their domain-specific requirements as some results depend on specific settings outside composition.

**Questions:**

Please refer to the weaknesses.

---

> ### Author Response · Authors · 2024-11-25
> **Response (Part 1/3)**
>
> Thank you for your comments and constructive feedback. We appreciate your recognizing our experimental contributions and that our codebase will be useful to the community. We have carefully addressed your concerns, adding three new sets of experiments and analyses based on your suggestions.
>
> We hope that our response and new experiments address your concerns, and we kindly ask you to consider raising your score. If, after reviewing our responses, you still have concerns about our work, we would greatly appreciate further guidance on how we can improve our work to achieve a better score in your review. Thank you!
> ### W1: Comparing more base models
> We agree that understanding how composability varies before and after post-training is an interesting question. While our framework indeed applies here, there are too many open questions here to answer in just one work. But to begin probing this question, we have taken your suggestion and run additional experiments on the instruction-tuned Llama-3-8b-instruct using the most-composable representative interventions from each category: MEMIT (editing), WANDA (compression), and RMU (unlearning). **In summary, these preliminary results suggest that our main findings hold between pretraining and instruction tuning**. These results suggest that factors such as architecture and pretraining regime are more likely to influence composability than post-training.
>
> **Compression → Editing is Best**: This observation still holds when pairing MEMIT (editing) and WANDA (compression), though the instruct model is robust to compression when it comes to editing performance. For the instruct model, the edit success for MEMIT→WANDA is 93.93% compared to 99.10% for WANDA→MEMIT.
>
> | Model | Intervention 1 | Intervention 2 | MMLU | Edit Success | Edit Generalization | Edit Locality |
> |--------|----------------|----------------|-------|--------------|-------------------|--------------|
> | Base | MEMIT | WANDA | 58.89% | 70.11% | 68.81% | 3.05% |
> | Instruct | MEMIT | WANDA | 61.12% | 93.93% | 84.47% | 2.69% |
> | Base | WANDA | MEMIT | 59.22% | 94.59% | 89.61% | 3.44% |
> | Instruct | WANDA | MEMIT | 61.47% | 99.10% | 84.77% | 3.26% |
>
> **Unlearning → Compression is Best**: This observation still holds when pairing RMU (unlearning) and WANDA (compression), with performance being relatively unchanged between the base and instruct versions. RMU→WANDA WMDP performance (lower is better) is  28.54% and 26.82% for the base and instruct models respectively compared to 31.86% and 31.83% for WANDA→RMU.
>
> | Model | Intervention 1 | Intervention 2 | MMLU | Avg WMDP |
> |--------|----------------|----------------|-------|-----------|
> | Base | RMU | WANDA | 55.64% | 28.54% |
> | Instruct | RMU | WANDA | 55.66% | 26.82% |
> | Base | WANDA | RMU | 57.12% | 31.86% |
> | Instruct | WANDA | RMU | 58.21% | 31.83% |
>
> **Editing←→Unlearning is Composable**: We continue to find that MEMIT and RMU are largely composable. However, the delta in performance between pairings is slightly higher for the instruct model for editing metrics.
>
> | Model | Intervention 1 | Intervention 2 | MMLU | Avg WMDP | Edit Success | Edit Generalization | Edit Locality |
> |-------|---------------|----------------|-------|-----------|--------------|-------------------|---------------|
> | Base | MEMIT | RMU | 56.19% | 29.28% | 95.62% | 89.38% | 3.48% |
> | Instruct | MEMIT | RMU | 54.59% | 27.28% | 93.88% | 87.00% | 3.19% |
> | Base | RMU | MEMIT | 55.94% | 30.14% | 96.87% | 92.97% | 3.40% |
> | Instruct | RMU | MEMIT | 53.99% | 25.53% | 89.41% | 84.44% | 3.43% |

---

> ### Author Response · Authors · 2024-11-25
> **Response (Part 2/3)**
>
> ### W2: Adding comparisons with different generations and sizes of the same model family.
> While we already study LMs from different families ranging from 7-9B parameters (Table 8), we agree further insights can be derived from considering model sizes and variations within model families. To address your suggestion, we have extended our experiments to add a new, smaller Llama model: Llama-3.2-1B. Our results are shown in the table below, where we observe that the smaller 1B Llama appears to have similar composability levels to its larger counterparts, as Llama-3.2 1B has an average order sensitivity of 3.60% whereas Llama-3 8B has 2.50%. We also observe that editing after compression is still better, though this effect is bigger in the 8B model. Results across parameter count and model family will be included in Section 4 of the paper.
>
> | Model Params | Composition | MMLU: Order Free Error | MMLU: Order Sensitivity | Edit Success: Order Free Error | Edit Success: Order Sensitivity | Edit Generalization: Order Free Error | Edit Generalization: Order Sensitivity | Edit Locality: Order Free Error | Edit Locality: Order Sensitivity |
> |:--|:--|--:|--:|--:|--:|--:|--:|--:|--:|
> | **8B** | LoRA↔WANDA | 40.00% | 0.00% | 13.00% | 8.00% | 47.00% | 1.00% | 97.00% | 1.00% |
> | **1B** | LoRA↔WANDA | 65.97% | 0.01% | 2.00% | 2.00% | 35.04% | 11.81% | 97.67% | 0.57% |
>
> | Model | Intervention 1 | Intervention 2 | MMLU | Edit Success | Edit Generalization | Edit Locality |
> |-------|---------------|----------------|------|--------------|-------------------|---------------|
> | 3-8B | LoRA | WANDA | 60.00% | 87.00% | 53.00% | 4.00% |
> | 3.2-1B | LoRA | WANDA | 34.03% | 98.00% | 64.96% | 2.33% |
> | 3-8B | FT | WANDA | 60.00% | 96.00% | 76.00% | 11.00% |
> | 3.2-1B | FT | WANDA | 35.01% | 90.26% | 75.06% | 1.49% |
> | 3-8B | WANDA | LoRA | 60.00% | 95.00% | 54.00% | 3.00% |
> | 3.2-1B | WANDA | LoRA | 34.03% | 100.00% | 76.76% | 2.90% |
> | 3-8B | WANDA | FT | 60.00% | 99.00% | 80.00% | 7.00% |
> | 3.2-1B | WANDA | FT | 33.82% | 100.00% | 65.26% | 1.77% |
>
> ### W3: Quantifying Results Sensitivity:
> Thanks for pointing this out—we have already addressed sensitivity but accidentally omitted these results. Thanks for pointing this out. Fortunately, our editing results are actually already averages over 10 data subsets. And all compression methods are deterministic. We will address your comment by adding standard errors into the editing results in Figure 2. For unlearning, we empirically observed they had tiny standard deviations when replicated (e.g., ±0.5% for accuracy), which we will also show in Figure 3. We have added the updated version of this table (Table 2 in the paper) including standard errors below. In this table, we notice the overall trends remain unchanged and not very sensitive to different data subsets.
>
>
> |  | **Edit Success**                                  |                           |                           |                           |                           |                           | **Edit Generalization**                                  |                           |                           |                           |                           |                           |  |
> |--------|----|---------------------------|---------------------------|---------------------------|---------------------------|---------------------------|---|---------------------------|---------------------------|---------------------------|---------------------------|---|--------|
> | Method  | **Order-free Error (↓)** | **Order-free Error (↓)** | **Order-free Error (↓)** | **Order Sensitivity (↓)** | **Order Sensitivity (↓)** | **Order Sensitivity (↓)** | **Order-free Error (↓)** | **Order-free Error (↓)** | **Order-free Error (↓)** | **Order Sensitivity (↓)** | **Order Sensitivity (↓)** | **Order Sensitivity (↓)** |        |
> |        | FT                       | MEMIT                   | LoRA                      | FT                        | MEMIT                    | LoRA                      | FT                       | MEMIT                   | LoRA                      | FT                        | MEMIT                    | LoRA                      |   # Wins     |
> | Wanda | 0.01 (0.01) | 0.05 (0.02) | 0.05 (0.01) | 0.03 (0.01) | 0.24 (0.03) | 0.08 (0.02) | 0.20 (0.02) | 0.10 (0.01) | 0.46 (0.03) | 0.04 (0.01) | 0.21 (0.03) | 0.01 (0.01) | 6.00 (0.00) |
> | SparseGPT | 0.01 (0.01) | 0.09 (0.02) | 0.14 (0.02) | 0.03 (0.01) | 0.04 (0.02) | 0.01 (0.01) | 0.23 (0.02) | 0.14 (0.02) | 0.48 (0.02) | 0.02 (0.01) | 0.03 (0.02) | 0.07 (0.02) | 4.00 (0.00) |
> | AWQ | 0.01 (0.01) | 0.07 (0.01) | 0.00 (0.01) | 0.04 (0.01) | 0.01 (0.01) | 0.08 (0.01) | 0.16 (0.02) | 0.11 (0.02) | 0.26 (0.03) | 0.05 (0.01) | 0.00 (0.01) | 0.14 (0.02) | 12.00 (0.00) |
> | GPTQ | 0.21 (0.02) | 0.16 (0.03) | 0.42 (0.03) | 0.36 (0.04) | 0.03 (0.03) | 0.37 (0.03) | 0.40 (0.04) | 0.20 (0.03) | 0.59 (0.04) | 0.36 (0.03) | 0.11 (0.02) | 0.27 (0.03) | 0.00 (0.00) |

---

> ### Author Response · Authors · 2024-11-25
> **Response (Part 3/3)**
>
> |  | **Strict Edit Locality**                                  |                           |                           |                           |                           |                           | **MMLU**                                  |                           |                           |                           |                           |                           |  |
> |--------|---------------------------------------------------|---------------------------|---------------------------|---------------------------|---------------------------|---------------------------|-----------------------------------------------------------|---------------------------|---------------------------|---------------------------|---------------------------|---------------------------|--------|
> | Method  | **Order-free Error (↓)** | **Order-free Error (↓)** | **Order-free Error (↓)** | **Order Sensitivity (↓)** | **Order Sensitivity (↓)** | **Order Sensitivity (↓)** | **Order-free Error (↓)** | **Order-free Error (↓)** | **Order-free Error (↓)** | **Order Sensitivity (↓)** | **Order Sensitivity (↓)** | **Order Sensitivity (↓)** |        |
> |        | FT                       | MEMIT                   | LoRA                      | FT                        | MEMIT                    | LoRA                      | FT                       | MEMIT                   | LoRA                      | FT                        | MEMIT                    | LoRA                      |   # Wins     |
> | AWQ | 0.86 (0.01) | 0.98 (0.00) | 0.91 (0.00) | 0.02 (0.01) | 0.00 (0.00) | 0.06 (0.01) | 0.41 (0.00) | 0.42 (0.00) | 0.41 (0.00) | 0.02 (0.00) | 0.01 (0.00) | 0.01 (0.00) | 0 |
> | Average | 0.89 (0.01) | 0.97 (0.00) | 0.95 (0.00) | 0.04 (0.00) | 0.01 (0.00) | 0.02 (0.00) | 0.40 (0.00) | 0.41 (0.00) | 0.41 (0.00) | 0.01 (0.00) | 0.01 (0.00) | 0.01 (0.00) | 0 |
> | GPTQ | 0.90 (0.01) | 0.97 (0.00) | 0.95 (0.00) | 0.08 (0.00) | 0.01 (0.00) | 0.01 (0.00) | 0.41 (0.00) | 0.42 (0.00) | 0.42 (0.00) | 0.01 (0.00) | 0.01 (0.00) | 0.01 (0.00) | 0 |
> | SparseGPT | 0.89 (0.01) | 0.96 (0.00) | 0.97 (0.00) | 0.03 (0.01) | 0.02 (0.00) | 0.01 (0.00) | 0.40 (0.00) | 0.41 (0.00) | 0.40 (0.00) | 0.00 (0.00) | 0.00 (0.00) | 0.00 (0.00) | 0 |
> | Wanda | 0.90 (0.01) | 0.97 (0.00) | 0.96 (0.00) | 0.05 (0.01) | 0.01 (0.00) | 0.01 (0.00) | 0.40 (0.00) | 0.41 (0.00) | 0.40 (0.00) | 0.00 (0.00) | 0.00 (0.00) | 0.00 (0.00) | 0 |
>
> ### W4: Domain-Specific Requirements:
> We believe this is not a weakness of our work, but a weakness of the popular models and interventions we consider—composability is currently worse than expected and less-predictable than we’d like. This is itself an important finding. So one of our key contributions is to encourage the community to address this problem, as it opens many new questions (like your listed W1 and W2). By already including results on popular models/interventions, our results can provide some initial guidance. But we do actually hope practitioners and researchers will now know to do such grid searches when necessary and share their results to continue fostering better, more-predictable composability.

---

> > ### Comment · Reviewer_KMdD · 2024-11-26
> >
> > I thank the authors for clarifying most of my concerns and so increased the rating accordingly.

---

### Author Response · Authors · 2024-11-25
**General Response**

Thank you to all reviewers for your thoughtful feedback! We appreciate the reviewers acknowledging  our contributions. Reviewers consistently applaud our extensive experiments, noting we provide a **“thorough documentation of interactions among various interventions”** [5FNv], which covers **“extensive experiments across 417 combinations of interventions”** [Mrdn]. Reviewers also appreciated our findings, noting our **“findings provide valuable insights into the compatibility and limitations of different intervention combinations, offering practical usage guidance”** [5FNv] and **“deeper insights”** [KMdD], how we **“present several important findings”** [1gkW],  and how our **“insights and findings are good”** [gvGR]. Reviewers emphasized the importance of our setting, noting our work studies an **“​​important problem in model updates and adaptations to real-world requirements”** [KMdD] and **“highlights an oversight in current evaluation methodologies”** [5FNv]. They also praise the clarity and readability of our paper, noting that it is **“lucid and easy to follow”** [KMdD] and **“very clear, nice, and easy to understand”** [gvGR].

Reviewers also raised several key points, which we address in point-by-point responses to each reviewer. Here in the general response, we briefly highlight new experiments added during this rebuttal period and address some important points raised by reviewers.

In response to reviewers’ comments, we conducted 4 new experiments to achieve the following:
- We study model size with a small, new Llama-3.2-1B, finding interventions are more composable on it than its larger variant
- We probe the effects of training objectives by adding an instruction-tuned Llama-3-8B-instruct, finding minimal impacts on composability
- With 12 new intervention combinations, we find that composing three interventions is straightforward and leaves MMLU performance surprisingly high.
- We study the robustness of our findings by reporting standard deviations for all feasible experiments, ultimately finding our results are quite robust.

These new experiments, along with our initial results, have strengthened our confidence in our contribution. Incorporating this feedback has improved the paper and we thank all reviewers for their helpful suggestions.

## General note to all reviewers
In our responses below, we abbreviate weaknesses by W# and questions by Q#.

*If you feel we have not sufficiently addressed your concerns to motivate increasing your score, we would love to hear from you further on what points of concern remain and how we can improve the work in your eyes. Thank you again!*

---

### Meta-Review · Area_Chair_r9WQ · 2024-12-23

**Metareview:**

This paper proposes a new framework to assess the effects of combining different test-time interventions on language models to improve factual accuracy, efficiency, and mitigate harmful outputs. The authors explore how these interventions interact when used together, analyzing the sensitivity to intervention order and the broader implications of composability. This research offers insights into the outcome of utilizing multiple interventions and emphasizes the need for multi-objective intervention metrics.

Most reviewers understood the paper and thought the results and the code to be a valuable contribution. Reviewers raised issues on if the findings will generalize broadly, and the statistical significance of some small differences. The main reject reviewer (5FNv) thought the analysis is not deep enough to chase down the root causes of failing composition but admits the author took a step is documenting where it is failing. Overall reviewers are consistent and did not identify critical flaws, so the decision follow their recommendation.

**Additional Comments On Reviewer Discussion:**

Authors attempted to address all issues and several reviewers acknowledged reading the responses.

---

### Decision · Program_Chairs · 2025-01-22

Accept (Poster)